

# Ozone and carbon monoxide observations over open oceans on R/V *Mirai* from 67° S to 75° N during 2012 to 2017: Testing global chemical reanalysis in terms of Arctic processes, low ozone levels at low latitudes, and pollution transport

Yugo Kanaya[1], Kazuyuki Miyazaki[1], Fumikazu Taketani[1], Takuma Miyakawa[1], Hisahiro Takashima[1,2], Yuichi Komazaki[1], Xiaole Pan[1,3], Saki Kato[2], Kengo Sudo[1,4], Jun Inoue[5], Kazutoshi Sato[6], and Kazuhiro Oshima[1,7]

[1] Japan Agency for Marine–Earth Science and Technology (JAMSTEC), Yokohama 2360001 Japan
[2] Fukuoka University, Fukuoka 8140133, Japan
[3] Now at Institute of Atmospheric Physics, Chinese Academy of Sciences, Beijing 100029, China
[4] Nagoya University, Nagoya 4648601, Japan
[5] National Institute of Polar Research, Tachikawa 1908518, Japan
[6] Kitami Institute of Technology, Kitami 0908507, Japan
[7] Now at Institute of Environmental Sciences, Rokkasho 0393212, Japan

*Correspondence to*: Yugo Kanaya (yugo@jamstec.go.jp)

**Abstract.** Constraints from ozone ($O_3$) observations over oceans are needed in addition to those from terrestrial regions to fully understand global tropospheric chemistry and its impact on the climate. Here, we provide a large data set of ozone and carbon monoxide (CO) levels observed (for 11 666 and 10 681 h, respectively) over oceans. The data set is derived from observations made during 24 research cruise legs of R/V *Mirai* during 2012 to 2017, in the Southern, Indian, Pacific, and Arctic Oceans, covering the region from 67° S to 75° N. The data are suitable for critical evaluation of the over-ocean distribution of ozone derived from chemical transport models. We first give an overview of the statistics in the data set and highlight key features in terms of geographical distribution and air mass type. We then use the data set to evaluate ozone concentration fields from Tropospheric Chemistry Reanalysis version 2 (TCR-2), produced by assimilating a suite of satellite observations of multiple species into a chemical transport model, namely CHASER. For long-range transport of polluted air masses from continents to the oceans, during which the effects of forest fires and fossil fuel combustion were recognized, TCR-2 gave an excellent performance in reproducing the observed temporal variations and photochemical buildup of $O_3$ when assessed from $\Delta O_3/\Delta CO$ ratios. For clean marine conditions with low and stable CO concentrations, two focused analyses were performed. The first was in the Arctic (>70° N) in September every year from 2013 to 2016; TCR-2 underpredicted $O_3$ levels by 6.7 ppb (21 %) on average. The observed vertical profiles from $O_3$ soundings from R/V *Mirai* during September 2014 had less steep vertical gradients at low altitudes (>850 hPa) than those obtained TCR-2. This suggests the possibilities of more efficient descent of the $O_3$-rich air from above or less efficient dry deposition on the surface than were assumed in the model. In the second analysis, over the western Pacific equatorial region (125–165° E, 10° S to 25° N), the observed $O_3$ level frequently decreased to less than 10 ppb in comparison to that obtained with TCR-2, and



also those obtained in most of the Atmospheric Chemistry Climate Model Intercomparison Project (ACCMIP) model runs for the decade from 2000. These results imply loss processes that are unaccounted for in the models. We found that the model's positive bias positively correlated with the daytime residence times of air masses over a particular grid, namely 165–180° E and 15–30° N; an additional loss rate of 0.25 ppb h$^{-1}$ in the grid best explained the gap. Halogen chemistry, which is commonly omitted from currently used models, might be active in this region and could have contributed to additional losses. Our open data set covering wide ocean regions is complementary to the Tropospheric Ozone Assessment Report data set, which basically comprises ground-based observations, and enables a fully global study of the behavior of O$_3$.

## 1 Introduction

The global burden and distribution of tropospheric ozone (O$_3$) have changed from preindustrial times to the present, and have induced radiative forcing of +0.4 W m$^{-2}$ (IPCC 2013) by interactions with the Earth's radiative field. The distribution of O$_3$ is critical for determining concentration fields of hydroxyl radicals, which control the lifetimes of many important chemical species, including methane. Changes in atmospheric chemistry and their impacts on the climate are often investigated by using O$_3$ distributions derived from chemical transport model simulations. Therefore the model performance determines the accuracy of the assessment, requiring model evaluation against field observations of levels of O$_3$ and its precursors. For example, the chemistry–climate models included in the Atmospheric Chemistry Climate Model Intercomparison Project (ACCMIP; Lamarque et al., 2013; Shindell et al., 2011) and the Chemistry–Climate Model Initiative (CCMI; Morgenstein et al., 2017) were carefully evaluated against field observations (e.g., Tilmes et al., 2016). Recently, an unprecedented comprehensive data set of O$_3$ measurements was systematically compiled under a community-wide activity, i.e., the Tropospheric Ozone Assessment Report (TOAR) (Cooper et al., 2014; Schultz et al., 2017), and this provided additional constraints for model simulations.

However, even with the data in the TOAR, observation data coverage over oceans is still poor; Schultz et al. (2017) reported that the true oceanic sites covered by TOAR were American Samoa (Global Atmosphere Watch, GAW), Sable Island (Canada National Air Pollution Surveillance), the Ieodo Ocean Research Station (Korea), Ogasawara (EANET), and Minamitorishima (GAW); Cape Grim, Amsterdam Island, and Mace Head were also included in other categories. For assessment of the Arctic region (AMAP, 2015), observations obtained truly over the Arctic Ocean were largely unavailable; only observations at costal or inland sites such as Alert, Barrow, Zeppelin, Pallas–Sodankyla, Summit, and Thule were used to test simulations. Measurements from individual cruises (e.g., Boylan et al., 2015; Dickerson et al., 1999; Kobayashi et al., 2008; Prados-Roman et al., 2015), O$_3$ soundings from the SHADOZ network (e.g., Thompson et al., 2017; Oltmans et al., 2001), and those from various campaigns (e.g., Kley et al., 1996; Takashima et al., 2008; Rex et al., 2014) have provided important O$_3$ data over oceanic regions. However, their spatio-temporal coverage over open oceans is too sparse to complete the global picture. Aircraft observations (e.g., HIAPER Pole-to-Pole Observations (HIPPO); Wofsy et al., 2011) also



sampled marine boundary layer but the measurement frequency was not necessarily high. One mature example of such global observations that include the over-ocean atmosphere is that for $CO_2$. For example, observational data at a large number of remote islands are available from the World Data Centre for Greenhouse Gases (WDCGG) database of the World Meteorological Organization (WMO)/GAW. Atmospheric $CO_2$ measurements are now accepted by SOCAT Version 4

(https://www.socat.info/), which is a database of ship-based observations, e.g., from research vessels and voluntary ships. Similarly dense $O_3$ observations over oceans are needed.

An understanding of the processes behind the $O_3$ concentration distribution is important. Young et al. (2018) reported that the rates of chemical processes (production and loss) and deposition/stratosphere–troposphere exchange could differ among models by factors of 2–3. More observational constraints for characterizing photochemical buildup and long-range transport

events using tracers such as carbon monoxide (CO) are required. The chemical loss term under clean marine conditions should also be examined; the loss rate caused by halogen chemistry, and the regions where such chemistry is important, need to be evaluated.

Since 2010, we have conducted ship-borne observations on R/V *Mirai* of the atmospheric composition, including the $O_3$ and CO levels, for more than 10 000 h. The geographical coverage was wide, namely the Arctic, Pacific, Indian, and

Southern Oceans. Such observations, together with currently available data sets, will enable critical testing of model simulations that cover the entire globe. In this paper, we present our observational data set for $O_3$ and CO for the first time. The data were separately analyzed for cases affected by long-range transport and those under clean marine conditions. For each case, the observations were compared with independent reanalysis data from Tropospheric Chemistry Reanalysis version 2 (TCR-2). The aims are to interpret the observations and to evaluate the reanalysis data over the oceans. The

reanalysis data were produced by assimilating a suite of satellite data for $O_3$ and precursors into a global chemical transport model, namely CHASER. The precursor emissions were simultaneously optimized. This has advantages over forward model simulations incorporating a bottom-up emission inventory because realistic emissions are taken into account, even those for recent years, for which a bottom-up emission inventory is not yet ready. TCR-2 was updated from the previous version TCR-1 (Miyazaki et al., 2015); the spatial resolution has been improved and newer satellite products are used for assimilation. For

TCR-1, evaluation against surface, sonde, and aircraft observations was successful (Miyazaki et al., 2015; Miyazaki and Bowman, 2017). For TCR-2, the performance has been evaluated using the KORUS-AQ aircraft campaign measurements over East Asia (Miyazaki et al., 2018). However, insufficient evaluation against data over remote oceans has been performed and our motivation in this study was to attempt this.

In Sect. 2, field observations and the assimilation model are outlined. In Sect. 3, geographical and statistical overviews of

the observational data are presented and then compared with the data from TCR-2. CO data and backward trajectories are used to classify $O_3$ data into cases that are influenced by long-range transport of pollution from continents and other cases, namely clean remote air masses. For the former cases, the reproducibility of the $O_3$ concentration levels and whether the chemical buildup is well reproduced by the reanalysis were tested for more than 20 events. For the latter cases, a particular focus was placed on underestimation for the Arctic Ocean and overestimation for the western Pacific equatorial region by



TCR-2. Similar trends to those for model members participating in the ACCMIP were observed. Possible explanations for these discrepancies were investigated.

## 2 Methodology

### 2.1 Observations on R/V *Mirai*

Atmospheric composition observations were conducted on R/V *Mirai* (8706 gross tons) of the Japan Agency for Marine–Earth Science and Technology (JAMSTEC) from 2010. The $O_3$ and CO levels were determined by UV and IR absorption methods (Models 49C and 48C, Thermo Scientific, Waltham, MA, USA), respectively. The instruments were located in an observational room on the top floor. Two Teflon tubes (6.35 mm o.d.) of length ~20 m were used to sample air near the bow to best avoid contamination from the ship's exhaust. The exhaust effect was clearly discerned in the 1 min $O_3$ data record as

high concentrations of NO in the exhaust titrated $O_3$. Minute data exceeding 3-sigma of the standard deviation in an hour were eliminated before producing hourly averages. The CO data for the same minutes were removed. The CO instrument alternately measured the ambient (for 40 min) and zero (for 20 min) levels. For the zero-level observations, CO was removed from the ambient air by using a zero-air generator equipped with a heated Pt catalyst (Model 96, Nippon Thermo, Uji, Japan). The $O_3$ instrument was calibrated twice per year in the laboratory, before and after deployment, using a primary standard $O_3$

generator (Model 49PS, Thermo Scientific, Waltham, MA, USA). The CO instrument was calibrated on board twice per year, on embarking and disembarking of the instrument, using a premixed standard gas ($CO/N_2$, 1.02 ppm, Taiyo-Nissan, Tokyo, Japan). The reproducibility of the calibration was to within 1 % for $O_3$ and 3 % for CO.

   The 24 cruise legs during which the two instruments were operated are listed in Table 1. During MR12-02 Legs 1 and 2, the CO instrument did not work well and only the $O_3$ data were used for analysis. The cruise regions ranged widely, from the

Arctic, the North, equatorial, and South Pacific Ocean, the eastern part of the Indian Ocean, to the Southern Ocean. The Arctic cruises took place every year during the period 2013 to 2016 (specifically during the MR13-06, 14-05, 15-03, and 16-06 cruises). Other cruises aimed to study geology, meteorology, and oceanography, and took place in the Pacific, Indian, and Southern Oceans. The western Pacific equatorial region and Indian Oceans were also frequently visited for operation of the TRITON buoy. Regions near Japan were frequently observed because the departure/arrival ports were often in that country.

Our data basically did not include observations made while the vessel was anchored in ports; exceptions were the inclusion of short on-port data when the ports were visited during cruise legs (see the far-right-hand column in Table 1). During many cruises, other instruments, i.e., for performing BC and fluorescent aerosol measurements, and multi-axis differential optical absorption spectroscopy (MAX-DOAS), were operated together (e.g., Taketani et al., 2016; Takashima et al., 2016). These data will be reported in future publications.



## 2.2 Reanalysis and chemistry climate models

The first version of tropospheric chemistry reanalysis from JAMSTEC, i.e., TCR-1, which used an ensemble Kalman filter (EnKF) approach with a global chemical transport model (CHASER) as a base forward model, has been previously described (Miyazaki et al., 2015). Here, the updated version, TCR-2, was used (Miyazaki et al., in preparation, 2018,

https://ebcrpa.jamstec.go.jp /tcr2/about_data.html). The detailed description of the basic data assimilation framework and the evaluation results using the KORUS-AQ aircraft measurements over East Asia are available in Miyazaki et al. (2018), while detailed global evaluations are ongoing (Miyazaki et al., in preparation, 2018). The major aspects of the update of TCR-1 were a finer horizontal resolution (1.1°, compared with 2.8° in TCR-1), assimilation of newer satellite data products [OMI $NO_2$ (QA4ECV), GOME-2 $NO_2$ (TM4NO2A v2.3), TES $O_3$ (v6), MOPITT CO (v7 NIR), and MLS $O_3$, $NO_3$ (v4.2)], and

extension of the period to 2017 (from 2005). A priori emissions were obtained from EDGAR v4.2 for anthropogenic sources, GFED v3.1 for open-fire emissions, and GEIA for biogenic sources. In addition to concentration fields of chemical species, $NO_x$ emissions (surface and lightning, separately) and CO were included in the state vector and were simultaneously optimized. An advantage was that analysis for recent years (e.g., 2017) was possible before development of a bottom-up emission inventory.

We also used the monthly ACCMIP ensemble simulations for the present day (Shindell et al., 2011), which were represented as simulation results for a decade from 2000 (Stevenson et al., 2013; Young et al., 2013). The monthly mean $O_3$ field from an ensemble member, MIROC-Chem, with a modeling framework similar to that of TCR-2, provided corresponding climatological concentration levels at a given location. These values were used to evaluate the performance of TCR-2, which uses meteorological data and estimated emissions in those particular years. Seven other ACCMIP ensemble

members that provide hourly $O_3$ concentrations at the Earth's surface (CESM-CAM-super fast, CMAM, GEOSCCM, GFDL-AM3, GISS-E2-R, MOCAGE, and UM-CAM) were also used for comparative analysis, in terms of frequency distribution of $O_3$ concentrations in the Arctic region (domain 1, 72.5–77.5° N, 190–205° E) and in a narrow western Pacific equatorial region (domain 3, 0–15° N, 150–165° E). The ACCMIP results were chosen for the latest intercomparison exercises whose data were publicly available.

## 2.3 Backward trajectory

Five-day backward trajectories from an altitude of 500 m above sea level (a.s.l.) were calculated every hour from the position of R/V *Mirai* by using NOAA's Hybrid Single-Particle Lagrangian Integrated Trajectory (HYSPLIT) model (Draxler and Rolph, 2013) to trace the origin areas of the observed air masses. GDAS1 three-dimensional meteorological field data with a resolution of 1.0° were used. Cases that did not involve traveling over land regions (at altitudes lower than

2500 m a.s.l.) during the five days were extracted as marine air mass cases. Here, the landmask data from NASA (https://ldas.gsfc.nasa.gov/gldas/data/0.25deg/landmask_mod44w_025.asc) at a resolution of 0.25° were used for making judgments.



## 3 Results and Discussion

### 3.1 Overview of geographical distributions of $O_3$ and CO

Figure 1a shows the entire $O_3$ data set from observations on R/V *Mirai* from 2012 to 2017. The covered latitudinal range was wide, from 67.00° S (at 06:00 UTC on 16 February 2017 during MR16-09 Leg 3) to 75.12° N (at 06:00 UTC on 6 September 2014 during MR14-05). The highest hourly concentration, namely 66.6 ppb, was recorded twice; first at 35.89° N, 141.95° E, about 70 km east off the coast of the Kanto area, Japan, at 04:00 UTC on 6 July 2012, and secondly at 32.29° N, 146.04° E, about 500 km southeast of the Kanto area at 14:00 UTC on 18 March 2014. During 57 h, the hourly values exceeded 60 ppb (which corresponds to the environmental standard in Japan) in a similar region off the coast of Japan, under the influence of Asian regional air pollution. The lowest hourly concentration, namely 4.3 ppb, was recorded twice; first at 5.86° N, 156.00° E and secondly at 7.96° N, 156.02° E at 09:00 and 21:00 UTC, respectively, on 11 March 2014. Values lower than 10 ppb were recorded for a total of 800 h; this will be discussed in the following section.

Figure 2a shows the entire CO data set. The highest concentration was 556 ppb, which was recorded at 58.00° N, 179.23° E, at 01:00 UTC on 26 September 2016 during the MR16-06 cruise, where a dense plume from severe forest fires in Russia reached as far as the Bering Sea. All 8 h in which the concentration exceeded 500 ppb were recorded on the same day. There were 59 other hours when the CO concentration exceeded 300 ppb, when anthropogenic emissions from East Asia and/or biomass burning were important.

Figure 3 shows an example of $O_3$ variations for data from MR14-06 Leg 1 with backward trajectories. The vessel started from Mutsu port (41.37° N, 141.24° E), stayed for about 32 h at Yokohama port (35.45° N, 139.66° E), and then headed to the western Pacific equatorial region. A latitudinal gradient was apparent; north of 27° N was dominated by air masses originating from the Asian continent, as shown by violet trajectory lines. During some hours, high levels, i.e., greater than 50 ppb of $O_3$, were recorded (points with red trajectory lines). In contrast, marine air masses from the east were dominant south of 27° N and the concentration levels decreased to less than 30 ppb. South of 15° N, even lower levels were dominant, i.e., <15 ppb. In equatorial regions, levels less than 10 ppb of $O_3$ were frequently observed. The latitudinal gradient, air mass exchange, and transport of photochemically produced $O_3$ are the three important factors determining distributions. Figure S1 shows the spatial distributions of $O_3$ and backward trajectories for each cruise leg, and their time series. All three factors listed above for the MR14-06 Leg 1 had major effects on the variations. The classification of air masses as marine and other cases was satisfactory for identifying cases of long-range transport of pollutants from continents, although some events with continental effects with traveling times longer than 120 h were probably wrongly categorized. Because longer trajectories may not be reliable, we will use this criterion (i.e., 120 h) in the following sections with a notice of such limitations. Stratospheric influences were not identified; we did not find any events with reasonable $O_3$-level enhancement accompanied by descent of air masses from 8 km or higher altitudes within 72 h prior to observations.



### 3.2 Comparisons between observations and TCR-2 data: Global features

Figures 1b and 2b show the geographical distributions of $O_3$ and CO obtained from TCR-2 along the cruise track. The nearest grid and time (2 h resolution) at the lowest layer were sampled to achieve the best possible comparison with the observations shown in Figs. 1a and 2a. Most features were in agreement with the observations, in terms of areas with high

concentrations and latitudinal gradients. This comparison indicates a reasonably high quality of reanalysis data over the oceans. This is partly because the $NO_x$ and CO emissions rates were optimally estimated during the data assimilation cycle and were reflected in the concentration field. The emission changes provide substantial influences on ozone forecasts over many regions. Assimilation of the TES ozone measurements has limited impacts on near surface ozone owing to its weak sensitivity to the lower troposphere.

In Fig. S1, the comparisons are shown as time-series plots. Excellent matches in the evolution with time of $O_3$ and CO concentrations were found for the latter period of MR14-02, i.e., during the period 12–22 March 2014, in the time series (Fig. S1g). The blue lines show plots of monthly climatological mean concentrations, which were obtained from MIROC-Chem. Although the monthly means sometimes followed the observed baseline trend for $O_3$, the performance of TCR-2 was superior, particularly for reproducing detailed peak patterns for CO. This case will be analyzed in Sect. 3.3.1 on

photochemical buildup. A similarly excellent performance was obtained for MR15-05 Leg 2 (Fig. S1r) and MR16-09 Leg 4 (Fig. S1w), when the ship returned to Japan from the south.

Clear gaps were also sometimes observed. These are probably related to limited reproducibility in the meteorological field and errors in the emission estimation and other physicochemical processes that were taken into account in the model system. A large gap was observed during the latter half of the MR14-01 cruise (Fig. S1f); the predictions produced by TCR-2 for CO

and $O_3$ in the eastern Indian Ocean were too high. The climatological mean from MIROC-Chem reproduced the observed level better, which suggests that false information was introduced from satellite observations to the surface concentrations, for instance, associated with the use of total column measurements and errors in vertical transport, or that the position of ITCZ was too far south in the simulation. During other cruises, smaller gaps in the peak occurrence times and peak concentration levels were found. For example, during MR14-04 Leg 2 (Fig. S1i), the peak times for $O_3$ and CO during the

period 27 July to 6 August 2014 were out of phase by about one to two days. This was probably caused by a small displacement of pollution air masses after long-range transport. High $O_3$ concentrations, i.e., >70 ppb (off the scale in the time-series plot) for 23 h were predicted by TCR-2, which was an overestimation compared with the observed concentrations. Except for 2 h on 9 October 2014, these were always in July (in 2012, 2013, and 2014). All were found within 200 km east of Japan's main islands, suggesting overestimation of photochemical $O_3$ production.

With such case-dependent reproducibility in mind, Fig. 1c summarizes the overall geographical distribution of the reanalysis/observation concentration ratios. For clarity, the data were limited to cases of marine origins, where the differences between the TCR-predicted and observed CO concentrations were less than 50 ppb ($N = 5662$), in contrast to Fig. 1a and b, where all data ($N = 11\,666$) were included. The model's underestimation in the Arctic Ocean and overestimation



over the oceans south of Japan (western Pacific equatorial or subtropical regions) are clearly shown. These features will be discussed in Sect. 3.3.

Figure 4 shows the latitudinal distribution of the observed and TCR-2 $O_3$ concentrations and their ratios. In Fig. 4a, the data were again limited to cases of marine origins, where the differences between the model-derived and observed CO concentrations were less than 50 ppb ($N$ = 5662). The variability shown here reflects seasonality and cases of long-range transport from continents over 120 h. The range of TCR-2/observation ratios (shown in red in Fig. 4b) narrowed when TCR-2 successfully captured the variability. Data selection ($N$ = 5662) also contributed to narrowing of the range (grey points indicate without data selection). The ratios binned by latitudes indicate that TCR-2 tended to give underestimations in high northern latitudes and overestimations in low latitudes. The 10th, 50th (median), and 90th percentiles (range shown as bars) of the ratios at high latitudes (>70° N) were 0.62, 0.81, and 0.94 ($N$ = 876); the range was off unity. The medians for low latitudes (0–10° N, 10–20° N, and 20–30° N) were 1.12 ($N$ = 1061), 1.28 ($N$ = 268), and 1.28 ($N$ = 195), respectively.

Figure 5 shows a scatterplot of the comparison. The data set was re-expanded by removing the criteria of marine air mass selection and CO differences (i.e., $N$ = 11 666) to show the overall correlation between the observed and TCR-2 data over the full range. The square of the correlation coefficient ($R^2$) was 0.59; this increased to 0.62 when clear outliers (latter half of MR14-01 and cases when TCR-2 exceeded 70 ppb) were omitted (red, $N$ = 11 305). The excellent performance of TCR-2 was assured by a resolution of 2 h in reproducing $O_3$ concentration variations. When the data were further binned to a resolution of one day (with observations for 6 h or more), $R^2$ improved to 0.66; this suggests that the original variability was partly derived from small shifts in the peak times by less than a day. In the past, reanalysis data were not used for such detailed diagnoses, but were normally compared with observations after monthly averaging or binning to coarse latitudinal bands. Recently, Akritidis et al. (2018) evaluated stratosphere-to-troposphere transport processes represented by the Copernicus Atmosphere Monitoring Service (CAMS) system, but evaluation of transport and transformation processes for tropospheric $O_3$ over oceans has not been reported with reanalysis. Here, a successful comparison was made for the first time at the daily or finer time-scale. The regression line for the data set, i.e., [TCR-2, ppb] = (6.90 ± 0.15) + (0.71 ± 0.01) × [obs, ppb], suggests a statistically significant positive intercept on the $y$-axis (i.e., TCR-2) and a slope of less than unity. The result was unchanged even when MR14-01 and cases with >70 ppb in TCR-2 were included. The green circles in Fig. 5 represent data from the Arctic region (>70° N), and clearly suggest an underestimation by TCR-2. In contrast, the blue circles from the western Pacific equatorial region (125–165° E, 10° S to 25° N) fell on the opposite side, with respect to the 1:1 line, indicating an overestimation by TCR-2. These two aspects will be discussed in detail in the following sections.

**3.3 Specific analysis with event or regional focuses: Implications for processes**

In this section, more detailed comparisons between observations and simulations are made, to shed light on the underlying mechanisms determining $O_3$ concentration levels and distributions.



### 3.3.1 Long-range transport events with photochemical production

We selected events with simultaneous increases in CO and $O_3$ levels or those with at least evident CO peaks. We then assessed the reproducibility of the peak concentrations and $\Delta O_3/\Delta CO$ ratios in TCR-2; the $\Delta O_3/\Delta CO$ ratio is an index of the efficiency of photochemical production of $O_3$ from CO as a precursor. In total, 23 cases were selected (Table 2 and magenta

rectangles in Fig. 2a). The two cases discussed below were studied in depth.

Figure 6 shows the time evolution of the $O_3$ and CO concentrations from observations and TCR-2 during the period 16–19 March 2014, when the vessel returned to Japan from the equatorial Pacific during MR14-02. The observational data along the entire track are shown, with the exact position of R/V *Mirai* at that time marked by a black circle. This part of the cruise has already been briefly mentioned in Sect. 3.2; here, the origins of pollution and the photochemical states for three episodes

with increased concentrations are discussed. First, at 09:00 UTC on 16 March 2014, at 23.69° N, 149.72° E, the CO concentration peaked at around 284 ppb. The geographical distribution of surface $O_3$ and CO in TCR-2 implied that this was a tongue-shaped pollution event originating from the Indochina peninsula and extending to the east; however, it was not. Instead, as inferred from the backward trajectories (Fig. S1g), weather charts, and time evolution of CO distributions from TCR-2 (Fig. 6), it should be interpreted as a belt of pollution originating from East Asia, present at the south edge of a high-

pressure system moving toward the southeast. An $O_3$ peak of 54.2 ppb occurred 10 h later, which was also captured by TCR-2. A second CO peak occurred the next day, at 19:00 UTC on 17 March 2014, at 28.56° N, 147.68° E; in this case an $O_3$ peak of 66.2 ppb occurred just 1 h before. TCR-2 suggested that the plume originated from East Asia and was transported to the east from the continent and then to the south, under the influence of another high-pressure system traveling over Japan and a low-pressure system that followed (see the 2nd row panel in Fig. 6 for 01:00 UTC on 17 March 2014). The timing and

location of the third peak were also well predicted; a CO peak of 273 ppb occurred at 17:00 UTC on 18 March 2014, at 32.88° N, 145.77° E, 3 h after the $O_3$ peak of 66.6 ppb (the highest in the data set). In this case, the air mass traveled directly from Central East China, with a large precursor emission (see the distribution of grids with high CO emissions in the panel for 01:00 UTC on 19 March 2014) via a quick westerly flow under the influence of passage of a cold front. The $\Delta O_3/\Delta CO$ ratio, calculated as the slope of the regression line in a scatterplot of hourly concentrations of $O_3$ and CO between 00:00

UTC on 17 March 2014 and 00:00 UTC 19 on March 2014, was $0.11 \pm 0.01$ ppb/ppb ($R = 0.83$) for observations and $0.06 \pm 0.01$ ppb/ppb ($R = 0.75$) for TCR-2 (Table 2, case G). The underestimation of $O_3$ peaks by TCR-2 (maximum 56.3 ppb) is attributable to a combination of less efficient production of $O_3$ and underestimation of CO peaks by TCR-2 (maximum 219 ppb).

Figure 7 shows a similar time evolution during MR14-04 Leg 2, when a plume reached the central Pacific from the Asian

continent about 5000 km away. The $\Delta O_3/\Delta CO$ ratio from 00:00 UTC on 2 August 2014 to 00:00 UTC on 4 August 2014 was $0.11 \pm 0.01$ ppb/ppb ($R = 0.90$) for observations and $0.14 \pm 0.01$ ppb/ppb ($R = 0.93$) for TCR-2; the maximum CO and $O_3$ concentrations were 249 and 38.0 ppb, respectively, for observations, and 220 and 37.2 ppb, respectively, for TCR-2 (Table 2, case I). The pollution source was probably forest fires in the far east of Russia (shown by plus signs in the upper left panel



of Fig. 7) with a smaller contribution from anthropogenic emissions from East Asia. Specifically, on 26 July 2014, the head of a plume extending from the forest fires arrived at northern Japan (see Zhu et al., submitted, 2018 and references therein for details) and affected our ship observations. The main body of the plume arrived from the west and traveled with a migrating high-pressure system until 6 August 2014. The observed CO and $O_3$ concentrations decreased during the period 29 July to 1 August 2014, when the vessel was located at the center of a low-pressure system traveling to the east. After the low-pressure system had weakened, the plume was still present in its southern part and was pushed northeast under the influence of the high-pressure system in the south, which again affected the ship observations during 2 and 3 August 2014, in the middle of the North Pacific.

Table 2 summarizes the results from all 23 cases, including the two events discussed above (cases F and G from MR14-02 and cases H and I from MR14-04 Leg 2). The span of the study area was from the Indian Ocean in the southern hemisphere, and the central equatorial Pacific Ocean to the Bering Sea (see magenta rectangles in Fig. 2a). In most cases, the observed $\Delta O_3/\Delta CO$ slopes were positive, suggesting photochemical buildup of $O_3$ with emissions of CO. The ranges were well reproduced by TCR-2. It is interesting to note that negative slopes for $\Delta O_3/\Delta CO$ were found for at least three cases and that these were also well captured by TCR-2. These were the cases where very high CO levels were recorded (479, 358, and 556 ppb for cases N, S, and T, respectively). The strong negative correlation for case N was influenced by the low $O_3$ concentrations (11.2 ppb) recorded with high CO levels. When such irregular points were removed, the observed slope was $0.01 \pm 0.01$ ppb/ppb. For case S, a strong CO peak occurred in the south of Japan (~200 km off the coast) on 23 January 2016, without an $O_3$ increase (Fig. S1r). For case T, a plume from a Russian forest fire affected the observations over the Bering Sea (Fig. S1s). Relatively fresh air pollution from nearby ships or weak UV in January and February for cases N and S, and weak co-emission of $NO_x$ from forest fires for case T, could cause the observed negative slopes. Processes other than daytime photochemistry might also have been important.

Figure 8 shows the correlation between the observed and reanalysis $\Delta O_3/\Delta CO$ ratios. The reasonable correlation, with a slope of $1.15 \pm 0.29$ and $R^2 = 0.42$, suggests that TCR-2 reproduced the state of $O_3$ production fairly well for various cases in different geographical domains in different seasons. Case B is an outlier in terms of correlation; observations and TCR-2 yielded $\Delta O_3/\Delta CO$ ratios of $0.28 \pm 0.03$ and $-0.12 \pm 0.06$ ppb/ppb, respectively. This was recorded at the center of the Bering Sea, where TCR-2 failed to reproduce the coinciding increases in the observed $O_3$ and CO concentrations.

The observed and simulated ranges of the $\Delta O_3/\Delta CO$ ratios are in accordance with previously reported data. On the Azores in the central Atlantic (38.73° N) in spring and on Sable Island (43.93° N) in Atlantic Canada in summer, the $\Delta O_3/\Delta CO$ ratios were uniform, at 0.3−0.4 ppb/ppb (Parrish et al., 1998). For TRACE-P aircraft measurements, Hsu et al. (2004) reported smaller values, i.e., 0.08 ppb/ppb in the tropics and 0.03 ppb/ppb in the extratropics, with CO > 200 ppb. Weiss-Penzias et al. (2006) reported 0.22 ppb/ppb in April and May 2004 for two long-range transport events that reached a mountain on the west coast of the United States from Asia. Tanimoto et al. (2008) summarized the range as being from slightly negative to ~0.4 ppb/ppb when observing Siberian fire plumes at Rishiri Island. Zhang et al. (2018) documented the $\Delta O_3/\Delta CO$ ratios at Pico Mountain in the Atlantic Ocean and found that the ratios for anthropogenic pollution were higher



(0.45–0.71 ppb/ppb) than those for observations affected by wildfires (0.12–0.71 ppb/ppb). They suggested that the low ratios from wildfires could be the result of lower $NO_x$/CO emission ratios compared with those for anthropogenic sources.

### 3.3.2 Arctic processes

In the Arctic Ocean (>70° N), the CO concentrations were regularly close to the background and stable (101 ± 10 ppb, Fig.

S1d, j, n, and s), suggesting that the measurements were not affected by strong pollution events. The average observed $O_3$ concentration was 31.3 ppb ($N = 1804$) and the reanalysis significantly underestimated this (24.6 ppb). The magnitude of the relationship was common for all four years of measurements (Fig. 9). A low bias with TCR-2 was also clear for cumulative frequency distributions of hourly $O_3$ concentrations from observations (black, $N = 1031$) and TCR-2 (red) in domain 1 [72.5–77.5° N and 190–205° E (155–170° W)] in September (Fig. 10). The frequency distributions for eight model members of

ACCMIP and their ensemble median (magenta) are also included in Fig. 10. The median was close to that for TCR-2, and similarly significantly underestimated, although three members (GEOSCCM, GFDL-AM3, and GISS-E2-R) showed better agreement with observations. These analyses suggest that in the simulations, the sources were too weak or the losses were too strong. The average diurnal variations were generally almost flat (the variability was within 5 % of the average) for observations and TCR-2 (not shown), suggesting that the missing processes did not show significant diurnal variability. At

high latitudes, the assimilated measurements have either low quality or low sensitivity in the troposphere, while the optimization of precursors emissions generally has limited impacts on ozone. The reanalysis ozone over the Arctic Ocean can be similar to the model predictions, unlike large increments obtained at low and mid latitudes, except when poleward transports are strong along low altitude.

During the MR14-05 cruise, 10 $O_3$ sondes were launched every other day at 22:00 UTC during the period 6–24 September

2014 (Inoue et al., 2018). The average vertical profile was compared with that from TCR-2 at the nearest grid (Fig. 11), to investigate the altitude to which this low bias of TCR-2 continued. At the lowest altitude near the surface, the underestimation by TCR-2 against the $O_3$ sonde observations was evident. However, the gap only continued to about 850 hPa, at which the two concentrations crossed over, and TCR-2 gave higher concentrations at higher altitudes. This suggests that the missing process was only important for the boundary layer, not for the entire troposphere. Inoue et al. (2018)

compared the $O_3$ sonde profiles with ERA-Interim (ERA-I) products, with a focus on troposphere–stratosphere exchange. They found that in the upper troposphere ERA-I had a high bias against observations, similarly to the case for TCR-2; this confirms that the underestimation by the model for the surface did not continue into the tropopause. One possibility is that the model's vertical (downward) mixing near the top of the boundary layer was too weak; this mixing would otherwise have effectively carried the $O_3$-rich air mass from higher altitudes. Another possibility is that dry deposition on the surface by the

model was too fast.

At Barrow (71.32° N, 156.6° W), the nearest ground-based site, the monthly average $O_3$ concentrations in September were 29.8 and 29.1 ppb, in 2013 and 2014, respectively. These values are close to our observations over oceans, and the model ensemble tended to underestimate concentrations in September (AMAP 2015), similarly to our case.



### 3.3.3 O₃ levels below 10 ppb at low latitudes

Figures 1c, 4, and 5 show that TCR-2 tended to overestimate $O_3$ concentrations over the western Pacific equatorial region. For the whole global data set, TCR-2 predicted $O_3$ levels below 10 ppb $O_3$ during 262 h, which is much less than the observed duration, i.e., 800 h. The occurrence frequencies in the region 125–165° E, 10° S to 25° N (defined as domain 2, $N$
= 2258) were large, namely 295 and 205 h for observations and TCR-2 predictions, respectively. When the study region was further narrowed to 150–165° E and 0–15° N (defined as domain 3, $N$ = 657), the discrepancy increased again, i.e., 199 and 80 h for observations and predictions, respectively. In March and December, observations were frequently conducted in domain 3 ($N$ = 211 and 321, respectively). Figure 12 shows the cumulative relative frequency distributions for observations, and the TCR-2 and ACCMIP models for these two months. In March (Fig. 12a), the observed $O_3$ concentrations in the low
ranges (<30th percentiles), and were as low as 5 ppb; these were overestimated by TCR-2. The shape of the distribution was in better agreement for higher ranges (30th to 100th percentiles). The ensemble medians for the ACCMIP runs were higher than the observations for any of the percentiles, whereas CESM-CAM-super fast, GISS-E2-R, GEOSCCM, and MOCAGE better captured features of the observed low concentrations (at higher percentiles GEOSCCM and MOCAGE gave too-high concentrations). For December (Fig. 12b), the performances of the models were poorer. Although CESM-CAM-super fast
and GISS-E2-R again captured the observed distributions in low ranges, all others, including the ACCMIP ensemble median and TCR-2, significantly overestimated the concentrations. The large variations among the model results could be the result of different assumptions regarding the dry deposition velocity of $O_3$. Because of the lack of direct constraints over the remote oceans on near surface concentrations in the current satellite observing systems, the systematic mismatches imply requirements for exploring model error sources to improve the reanalysis quality.

The average relative diurnal variations normalized to the maximum concentrations during these two months (Fig. 13, $x$-axis is UTC + 11 h, adjusted to local time for the selected region) showed a pattern of daytime decreases, suggesting photochemical destruction. However, the observed decrease was stronger (15 %) and earlier than those simulated in the TCR-2 and ACCMIP runs, in which the $HO_x$ cycle was primarily responsible for photochemical destruction. This feature may indicate $O_3$ loss via a different process, e.g., halogen chemistry, which is not included in any of the model simulations
considered in this study.

To obtain further insights into the regions in which this additional loss could be important, the differences between the $O_3$ levels predicted by TCR-2 and the observed levels were calculated and the correlations with the residence times (in the daytime) of back trajectories in 15°× 15° grid regions around domain 3 were examined. Use of the residence time during the day can be justified because the destruction probably occurred in the daytime (Fig. 13). Figure S2 shows the results for 17
regions, covering 120–195° E and 15° S to 45° N. We found that the residence time in the grid region 165–180° E and 15–30° N, located northeast of domain 3, had the largest positive correlation coefficient (Fig. 14). This suggests that this region is a possible "hot spot" for additional loss. It is worth noting that data from all five cruises contributed to the positive correlation, which suggests that the relationship is reproducible. The slope of the regression line was 0.25 ppb h⁻¹. This



corresponds to the possible loss rate in the grid that best explains the discrepancy between observations and TCR-2 predictions. If the rate is attributed to dry deposition on the sea surface, a deposition velocity as high as 0.33 cm s$^{-1}$ is required, assuming a boundary layer height of 1000 m and an average O$_3$ concentration of 17.4 ppb. This high velocity is not supported by previous studies (e.g., Ganzeveld et al., 2009; Hardacre et al., 2015). Such a loss rate can be more easily

explained by bromine and/or iodine chemistry in the atmosphere (e.g., Saiz-Lopez et al., 2012, 2014). We observed elevated IO radical concentrations with a MAX-DOAS instrument aboard R/V *Mirai* during the cruises included in the analysis in Fig. 14. Further analysis will be reported in a future publication (Takashima et al., in preparation, 2018).

Substantially reduced O$_3$ levels (<10 ppb) have been reported in the marine boundary layer over the equatorial Pacific. Johnson et al. (1990) found near-zero (3 ppb or less) O$_3$ concentrations in the central equatorial Pacific during April and May.

Kley et al. (1996) observed similar events in March 1993, with concentrations occasionally reaching 3–5 ppb, from O$_3$ soundings in a region between the Solomon Islands (9.4° S, 160.1° E) and Christmas Island (2° N, 157.5° W). Takashima et al. (2008) reported substantially reduced O$_3$ events throughout the year on Christmas Island. Rex et al. (2014) used a combination of O$_3$ sounding data from the TransBrom cruise of R/V *Sonne* in October 2009, chemical transport models, and satellite observations to identify a region with strong O$_3$ depletion in the marine boundary layer at 0–10° N along the north–

south cruise track (at around 150° E). This is in good agreement with our observations. Hu et al. (2010) reported that average O$_3$ concentrations at Kwajalein Island (8.72° N, 167.73° E), Republic of the Marshall Islands, located near our "hot spot" grid, during July, August, and September 1999, were lower than 10 ppb throughout 24 h, with an afternoon decrease of about 1 ppb. Gómez Martín et al. (2016) reported year-round continuous observations of surface O$_3$ at San Cristóbal (0.90° S, 89.61° W) and at Puerto Villamil, Isabela Island (0.96° S, 90.97° W), in the Galápagos Islands in the equatorial eastern

Pacific. Daily averages as low as 5–6 ppb were observed during the period February to May. During the Malaspina circumnavigation in 2010, O$_3$ concentrations as low as 3.4 ppb were detected around the central Pacific equatorial region (Prados-Roman et al., 2015). During PEM-West A in October 1991, Singh et al. (1996) reported O$_3$ levels as low as 8–9 ppb from aircraft measurements at altitudes of 0.3–0.5 km in the region 0–20° N in the western and central Pacific. It is worth noting that the O$_3$ levels over the Atlantic were typically higher, e.g., 30–35 ppb (Read et al., 2008), even when halogen-

mediated destruction was considered. In these studies, detailed statistical comparisons of the observed O$_3$ levels with chemistry-climate model simulations or reanalysis data as focused here have not been achieved. Conducting continuous observations of O$_3$ during a large number of cruise legs was useful for obtaining a large data set for performing detailed statistical analysis.

The influence of halogen (bromine and/or iodine) chemistry on O$_3$ levels has been studied using model simulations at

assumed (e.g., Davis et al., 1996; Hu et al., 2010) and observed levels of halogen compounds (Mahajan et al., 2012; Großmann et al., 2013; Dix et al., 2013; Prados-Roman et al., 2015; Koenig et al., 2017) over the Pacific. For example, for bromine chemistry, Hu et al. (2010) estimated a photochemical loss rate of up to 0.12 ppb h$^{-1}$. A loss of about 0.4 ppb d$^{-1}$ can be attributed to halogen chemistry in the marine boundary layer south of Hawaii (Dix et al., 2013). From CAM-Chem-based global model simulations, Saiz-Lopez et al. (2012) estimated an annually integrated rate of surface O$_3$ loss through



halogen chemistry as ~0.15 ppb h$^{-1}$ in the daytime, over the region 20° S to 20° N. On the basis of GEOS-Chem, Sherwen et al. (2016) suggested that halogen chemistry would reduce surface O$_3$ levels by ~3–5 ppb over our domain 3. These loss rates caused by halogen chemistry are in fair agreement with the required additional loss rate, i.e., ~0.25 ppb h$^{-1}$, estimated in this study.

Our observational O$_3$ data set with a large geographical coverage will be useful for evaluating up-to-date model simulations with inclusion of halogen chemistry. Analysis at mid-latitude regions over the Pacific are also of interest because the importance of halogen chemistry in this region has been indicated (e.g., Nagao et al. 1999; Galbally et al. 2000; Watanabe et al., 2005).

## 4 Summary and outlook

We compiled a large data set of shipborne in situ observations of O$_3$ and CO levels with a 1 h resolution, which were recorded on R/V *Mirai* over the Arctic, Bering, Pacific, Indian, and Southern Oceans from 67° S to 75° N, during the period 2012 to 2017. We used the data set to evaluate tropospheric chemistry reanalysis data from TCR-2 and ACCMIP model simulations. TCR-2 captured the basic features of the observations, including the latitudinal gradient and air mass exchange, and therefore enabled interpretation of observations regarding transport and pollution sources. Correlations with observations

were sufficient ($R^2$ up to 0.62 for hourly data and 0.67 for daily data). This suggests an excellent performance by TCR-2 in representing the temporal and geographical distributions of surface O$_3$. For over 23 long-range transport events with CO and/or O$_3$ buildup, variations in the $\Delta O_3/\Delta CO$ ratios were well reproduced by TCR-2. This suggests that the nature of photochemical evolution during transport of pollution plumes was also well captured. However, two major discrepancies were identified; in the Arctic (>70° N) in September, TCR-2 and the ensemble median of model runs of ACCMIP tended to

underestimate O$_3$ levels. From analysis of O$_3$ sonde measurements, we concluded that the gap was related to processes relevant to the boundary layer; downward mixing from the free troposphere, with a higher O$_3$ abundance, might have been stronger, or dry deposition on the sea surface might have been weaker, than assumed in the model. Conversely, in the western Pacific equatorial region, TCR-2 and ACCMIP simulations significantly overestimated the observed O$_3$ levels, which were often less than 10 ppb. The minimum in the observed diurnal pattern occurred earlier than those in the models.

The concentration gap correlated with the residence time of trajectories over a particular grid, i.e., 165–180° E and 15–30° N. These analyses indicate the importance of halogen chemistry, which is not accounted for in the models, and the region in which it is active. The data set from our observations (which will continue) is open and complements the TOAR data collection and will be useful for critically evaluating global-scale chemistry–climate and chemical transport model simulations, including those from CCMI, AerChemMIP (Collins et al., 2017), and later inter-comparisons.



## Data availability

The observational data set for $O_3$ and CO obtained on R/V *Mirai* for the study period (2012–2017) is collectively available from https://ebcrpa.jamstec.go.jp/atmoscomp/obsdata/ as a single text file and from http://www.godac.jamstec.go.jp/darwin/e for individual cruise legs. TCR-2 reanalysis data are available from http://ebcrpa.jamstec.go.jp/tcr2/download.html.

## Author contribution

YK (Kanaya) designed the study, conducted analyses, wrote the manuscript, managed the instruments and created the observational data set. KM and KS (Sudo) created TCR-2 data. FT, TM, HT, YK (Komazaki), XP and SK substantially contributed to shipborne observations and preparation. HT and SK also provided supporting data from MAX-DOAS observations. JI, KS (Sato) and KO conducted shipborne ozone soundings and provided the data. All co-authors provided professional comments to improve the manuscript.

## Competing interest

The authors declare that they have no conflict of interest.

## Acknowledgments

We gratefully acknowledge assistance from the Principal Investigators of all cruises and support from Global Ocean Development Inc. and Nippon Marine Enterprise, Ltd. This research was supported by the Coordination Funds for Promoting AeroSpace Utilization and by the Arctic Challenge for Sustainability (ArCS) project of the Ministry of Education, Culture, Sports, Science and Technology (MEXT), MEXT/JSPS KAKENHI Grant Numbers 24241009, 18H04143 and 18H01285, and the Environment Research and Technology Development Fund (2-1803) of the Ministry of the Environment, Japan. We thank Irina Petropavlovskikh (NOAA/ESRL) for providing data for Barrow. We thank Helen McPherson, PhD, from Edanz Group (www.edanzediting.com/ac) for editing a draft of this manuscript.

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



**Table 1.** Overview of cruise legs of R/V *Mirai* used in this study.

| Cruise | Study area | Departure | | Arrival | | Remark |
|---|---|---|---|---|---|---|
| MR12-02 Leg1 | Western North Pacific | 00:00UTC 4 Jun 2012 | Mutsu | 08:00UTC 24 Jun 2012 | Onahama (out of port) | |
| MR12-02 Leg2 | Western North Pacific | 08:00UTC 24 Jun 2012 | Onahama (out of port) | 00:00UTC 12 Jul 2012 | Mutsu | Hachinohe Port from 0030 to 0800 UTC Jul 11 (no data) |
| MR13-04 | Western North Pacific | 23:50UTC 9 Jul 2013 | Yokohama | 00:00UTC 29 Jul 2013 | Mutsu | |
| MR13-05 | Bering Sea | 23:50UTC 12 Aug 2013 | Mutsu | 17:40UTC 26 Aug 2013 | Dutch Harbor | |
| MR13-06 Leg1 | Arctic Ocean, Bering Sea | 18:00UTC 28 Aug 2013 | Dutch Harbor | 18:40UTC 7 Oct 2013 | Dutch Harbor | |
| MR13-06 Leg2 | Bering Sea, North Pacific | 17:40UTC 9 Oct 2013 | Dutch Harbor | 23:50UTC 20 Oct 2013 | Mutsu | |
| MR14-01 | East Indian Ocean, Equatorial Region | 23:00UTC 8 Jan 2014 | Mutsu | 00:00UTC 13 Feb 2014 | Palau | |
| MR14-02 | Western Pacific, Equatorial Region | 00:00UTC 15 Feb 2014 | Koror, Palau | 00:00UTC 23 Mar 2014 | Mutsu | Hachinohe Port, 0400-0900UTC 21 Mar 2014 (no data) |
| MR14-04 Leg1 | Western North Pacific | 22:10UTC 8 Jul 2014 | Yokohama | 04:00UTC 15 Jul 2014 | Kushiro | |
| MR14-04 Leg2 | North Pacific | 01:00UTC 17 Jul 2014 | Kushiro | 17:50UTC 29 Aug 2014 | Dutch Harbor | |
| MR14-05 | Arctic Ocean, Bering Sea, North Pacific | 18:10UTC 31 Aug 2014 | Dutch Harbor | 00:20UTC 10 Oct 2014 | Yokohama | |
| MR14-06 Leg1 | Western Pacific, Equatorial region | 06:10UTC 4 Nov 2014 | Mutsu | 23:20UTC 17 Dec 2014 | Chuuk | Yokohama Port, from 2310UTC 5 Nov to 0700UTC 7 Nov (with data) |
| MR14-06 Leg2 | Western Pacific Equatorial region | 00:07UTC 20 Dec 2014 | Chuuk | 00:10UTC 19 Jan 2015 | Palau | |
| MR14-06 Leg3 | Western Pacific, East Indian Ocean Equatorial Region | 00:00UTC 22 Jan 2015 | Palau | 00:00UTC 25 Feb 2015 | Mutsu | Hachinohe Port, from 2330UTC 23 Feb to 0700UTC 24 Feb (with data) |
| MR15-03 leg 1 | North Pacific, Bering Sea, Arctic Ocean | 22:50UTC 23 Aug 2015 | Mutsu | 18:50UTC 6 Oct 2015 | Dutch Harbor | |
| MR15-03 leg 2 | Bering Sea, North Pacific | 18:10UTC 9 Oct 2015 | Dutch Harbor | 23:50UTC 21 Oct 2015 | Mutsu | Hachinohe Port, from 2300UTC 20 Oct to 0650UTC 21 Oct (with data) |
| MR15-04 | Western Pacific, East Indian Ocean Equatorial Region | 06:00UTC 5 Nov 2015 | Mutsu | 02:20UTC 20 Dec 2015 | Jakarta | Hachinohe Port, from 2250UTC 5 Nov to 0620UTC 6 Nov (with data) |
| MR15-05 leg 1 | East Indian Ocean | 03:10UTC 23 Dec 2015 | Jakarta | 00:50UTC 11 Jan 2016 | Bali | |
| MR15-05 leg 2 | East Indian | 01:00UTC | Benoa, Bali | 23:50UTC | Yokohama | |





| | Ocean, Western North Pacific | 13 Jan 2016 | | 24 Jan 2016 | | |
|---|---|---|---|---|---|---|
| MR16-06 | Arctic Ocean, Bering Sea, North Pacific | 00:00UTC 22 Aug 2016 | Hachinohe | 00:00UTC 5 Oct 2016 | Mutsu | Nome port, 1600-2010UTC 23 Sep (with data); Hachinohe Port from 2230UTC 3 Oct to 0720UTC 4 Oct (with data) |
| MR16-08 | Western Pacific Equatorial Region | 07:00UTC 27 Nov 2016 | Shimizu | 21:00UTC 23 Dec 2016 | Suva | |
| MR16-09 leg 1 | South Pacific | 17:10UTC 26 Dec 2016 | Suva | 11:00UTC 17 Jan 2017 | Puerto Mont | |
| MR16-09 leg 3 | Southern Ocean | 13:10UTC 8 Feb 2017 | Punta Arenas | 21:00UTC 4 Mar 2017 | Auckland | |
| MR16-09 leg 4 | Western Pacific | 21:20UTC 7 Mar 2017 | Auckland | 00:00UTC 28 Mar 2017 | Mutsu | Hachinohe Port, from 2240UTC 26 Mar to 0650UTC 27 Mar (with data) |





**Table 2.** Cases of long-range transport events used for examination of photochemistry (unit for $\Delta O_3/\Delta CO$ is ppb/ppb).

| case | Start time | End time | Latitude (°N) | Longitude (°E) | $R$ (obs) | $\Delta O_3/\Delta CO$ (obs) | $R$ (TCR-2) | $\Delta O_3/\Delta CO$ (TCR-2) | COmax (obs) | COmax (TCR-2) | O₃max (obs) | O₃max (TCR-2) |
|---|---|---|---|---|---|---|---|---|---|---|---|---|
| A | 22:00UTC 15 Aug 2013 | 22:00UTC 17 Aug 2013 | 41.32–45.22 | 151.81–162.71 | 0.81 | 0.42±0.04 | 0.68 | 0.96±0.15 | 101.6 | 87.5 | 38.2 | 40.0 |
| B | 12:00UTC 23 Aug 2013 | 12:00UTC 25 Aug 2013 | 54.47–56.59 | 182.07–193.25 | 0.79 | 0.28±0.03 | -0.30 | -0.12±0.06 | 101.9 | 102.7 | 33.1 | 26.0 |
| C | 00:00 UTC 05 Oct 2013 | 22:00 UTC 06 Oct 2013 | 56.95–65.12 | 191.36–192.82 | 0.80 | 0.54±0.06 | 0.85 | 0.86±0.08 | 116.4 | 117.2 | 43.1 | 38.7 |
| D | 23:00 UTC 16 Oct 2013 | 23:00 UTC 18 Oct 2013 | 40.08–40.88 | 143.72–154.34 | 0.75 | 0.32±0.04 | 0.58 | 0.15±0.03 | 137.4 | 129.2 | 51.1 | 40.9 |
| E | 15:00 UTC 12 Jan 2014 | 15:00 UTC 14 Jan 2014 | 25.21–31.33 | 126.85–136.28 | 0.66 | 0.03±0.01 | 0.29 | 0.01±0.01 | 331.2 | 269.7 | 52.8 | 45.8 |
| F | 00:00 UTC 13 Mar 2014 | 00:00 UTC 15 Mar 2014 | 11.51–18.95 | 151.64–154.58 | 0.83 | 0.34±0.03 | 0.93 | 0.51±0.03 | 137.4 | 125.3 | 35.4 | 37.3 |
| G | 00:00 UTC 17 Mar 2014 | 00:00 UTC 19 Mar 2014 | 25.08–34.12 | 145.21–149.15 | 0.86 | 0.11±0.01 | 0.75 | 0.06±0.01 | 299.1 | 219.0 | 66.6 | 56.3 |
| H | 13:00 UTC 24 Jul 2014 | 10:00 UTC 26 Jul 2014 | 42.67–45.32 | 152.68–157.02 | 0.57 | 0.38±0.09 | 0.92 | 0.23±0.01 | 132.2 | 208.0 | 48.2 | 47.9 |
| I | 00:00 UTC 02 Aug 2014 | 00:00 UTC 04 Aug 2014 | 46.99–47.01 | 171.62–177.97 | 0.90 | 0.11±0.01 | 0.93 | 0.14±0.01 | 249.4 | 220.0 | 38.0 | 37.2 |
| J | 00:00 UTC 25 Aug 2014 | 00:00 UTC 27 Aug 2014 | 51.15–53.38 | 208.00–221.14 | 0.87 | 0.53±0.04 | 0.70 | 0.15±0.02 | 119.0 | 116.1 | 35.4 | 31.5 |
| K | 00:00 UTC 08 Nov 2014 | 00:00 UTC 10 Nov 2014 | 26.61–32.60 | 141.81–149.90 | 0.92 | 0.44±0.03 | 0.67 | 0.23±0.04 | 154.4 | 145.1 | 55.6 | 46.4 |
| L | 02:00 UTC 07 Feb 2015 | 21:00 UTC 08 Feb 2015 | -1.60–2.16 | 89.95–90.19 | 0.95 | 0.21±0.01 | 0.96 | 0.60±0.03 | 181.1 | 105.7 | 31.8 | 24.4 |
| M | 00:00 UTC 18 Feb 2015 | 00:00 UTC 20 Feb 2015 | 24.42–30.05 | 123.50–131.79 | 0.86 | 0.05±0.004 | 0.19 | 0.02±0.02 | 390.7 | 222.1 | 58.9 | 51.5 |
| N | 17:00 UTC 21 Feb 2015 | 17:00 UTC 23 Feb 2015 | 34.29–40.27 | 139.89–142.33 | -0.27 | -0.03±0.02 | -0.20 | -0.04±0.03 | 478.6 | 220.6 | 53.6 | 52.9 |
| O | 00:00 UTC 05 Oct 2015 | 17:00 UTC 06 Oct 2015 | 53.98–60.57 | 192.01–193.51 | 0.76 | 0.38±0.05 | 0.38 | 0.09±0.03 | 127.3 | 126.2 | 46.1 | 36.1 |
| P | 14:00 UTC 16 Nov 2015 | 12:00 UTC 18 Nov 2015 | -9.21–-0.19 | 114.13–119.13 | 0.82 | 0.29±0.03 | 0.83 | 0.37±0.04 | 148.8 | 185.0 | 28.9 | 48.7 |
| Q | 18:00 UTC 16 Dec 2015 | 15:00 UTC 18 Dec 2015 | -6.14–-4.04 | 101.01–102.27 | 0.39 | 0.09±0.04 | -0.75 | -0.03±0.005 | 94.7 | 214.4 | 18.8 | 17.5 |
| R | 04:00 UTC 27 Dec 2015 | 04:00 UTC 29 Dec 2015 | -24.78–-21.16 | 110.59–112.77 | 0.90 | 0.33±0.03 | 0.89 | 0.44±0.03 | 82.7 | 93.1 | 28.3 | 37.3 |
| S | 08:00 UTC 23 Jan 2016 | 20:00 UTC 24 Jan 2016 | 31.97–35.03 | 138.91–139.94 | -0.40 | -0.02±0.01 | -0.63 | -0.21±0.05 | 357.8 | 168.5 | 46.4 | 46.3 |
| T | 01:00 UTC 25 Sep 2016 | 01:00 UTC 27 Sep 2016 | 54.78–61.49 | 173.23–184.47 | -0.74 | -0.03±0.004 | -0.74 | -0.16±0.02 | 555.7 | 161.4 | 47.9 | 45.3 |
| U | 12:00 UTC 27 Sep 2016 | 12:00 UTC 29 Sep 2016 | 47.42–53.85 | 161.78–171.57 | 0.00 | 0.00±0.01 | -0.78 | -0.10±0.01 | 317.9 | 139.6 | 43.6 | 36.1 |
| V | 12:00 UTC 03 Dec 2016 | 10:00 UTC 05 Dec 2016 | 7.66–12.46 | 136.43–137.00 | 0.67 | 0.24±0.04 | 0.97 | 0.60±0.03 | 92.7 | 99.1 | 25.1 | 34.8 |
| W | 00:00 UTC 20 Mar 2017 | 00:00 UTC 22 Mar 2017 | 19.86–27.70 | 147.83–152.62 | 0.81 | 0.25±0.03 | 0.83 | 0.13±0.02 | 214.8 | 214.6 | 63.0 | 59.0 |





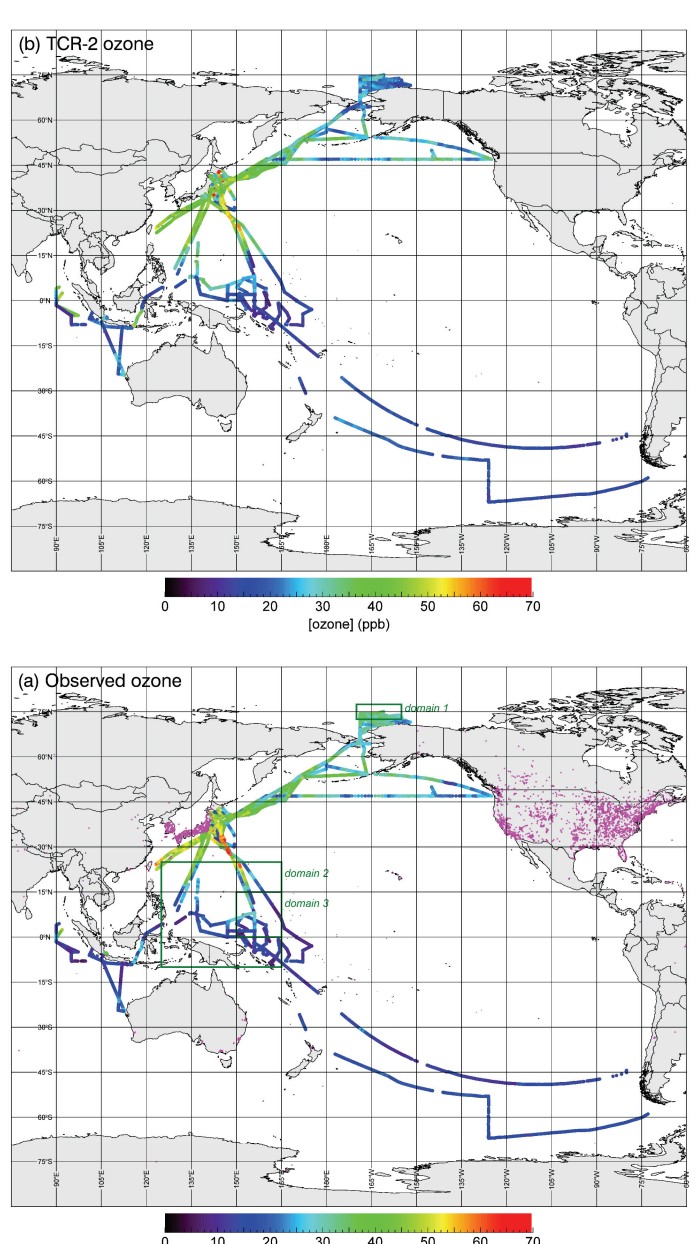

**Figure 1.** continued.





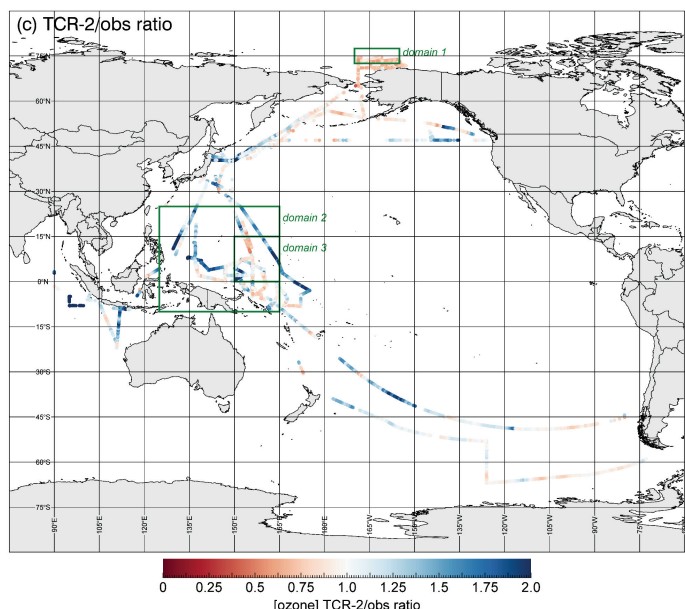

**Figure 1.** Geographical distributions from (a) observed hourly O$_3$ concentrations ($N = 11\,666$) on R/V *Mirai* during 24 research cruise legs in 2012–2017 and (b) those from reanalysis (TCR-2) of data along R/V *Mirai* cruise track; (c) reanalysis/observation ratios (TCR-2/obs) of O$_3$ concentrations. In (a) stationary points of TOAR data set are plotted (magenta dots). In (a) and (c), three green rectangles show focused domains in Arctic (domain 1, 72.5–77.5° N, 190–205° E) and two in western Pacific equatorial region (domain 2, 10° S to 25° N, 125–165° E; domain 3, 0–15° N, 150–165° E).




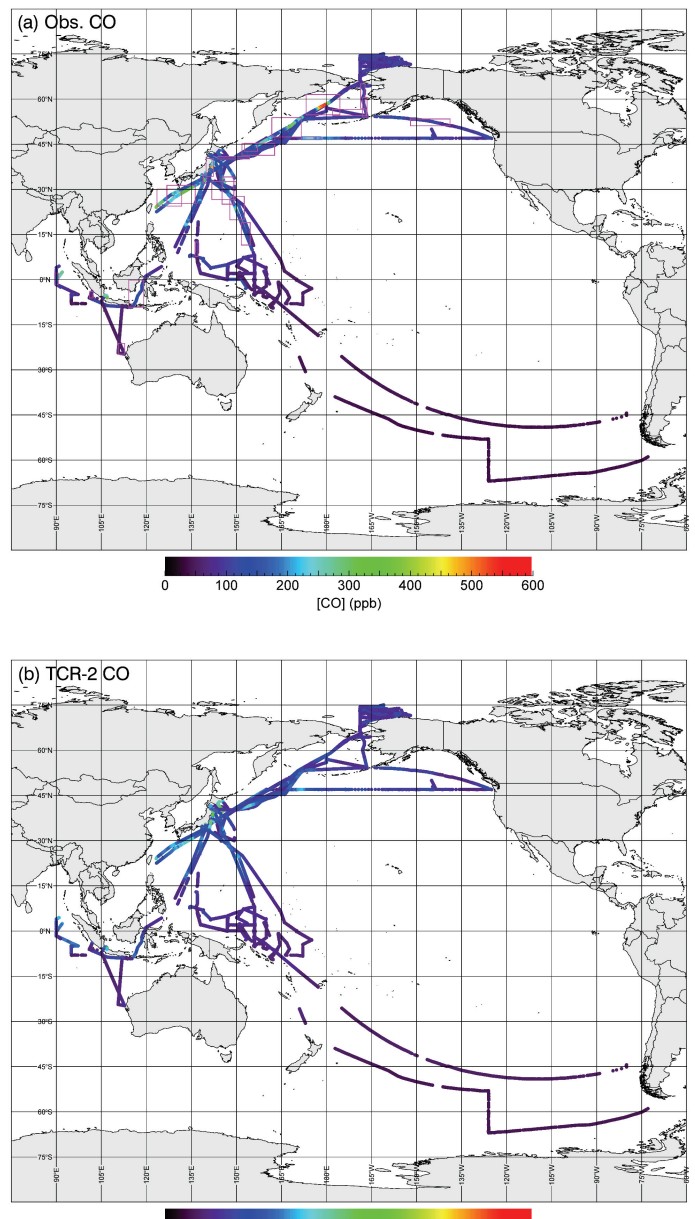

**Figure 2.** Geographical distributions of (a) observed CO concentrations and (b) those from TCR-2. In (a) magenta boxes indicate 23 regions where ΔO₃/ΔCO ratios were analyzed (see Fig. 8 and Table 2).





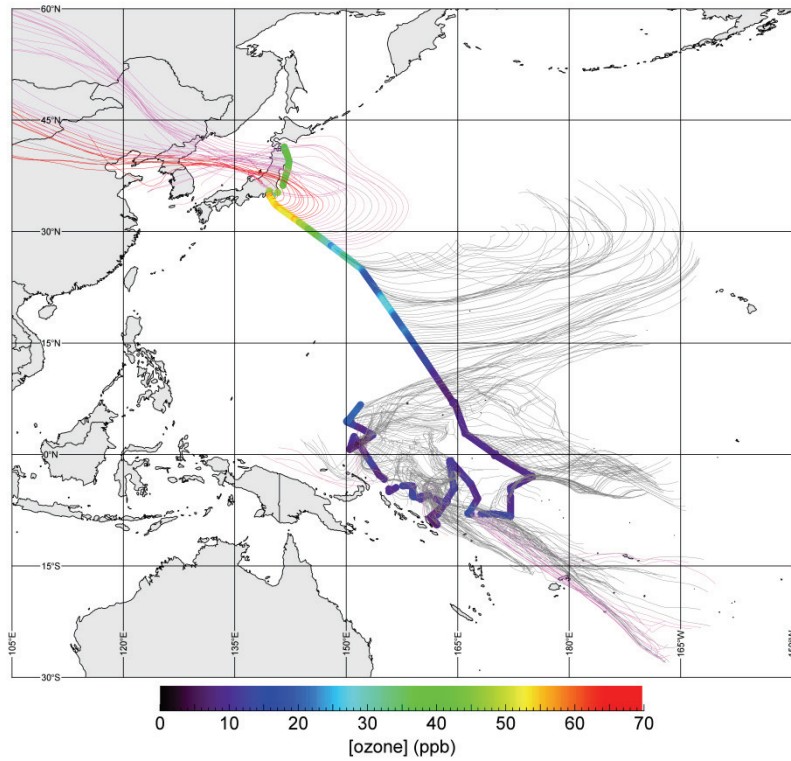

**Figure 3.** Observed O$_3$ concentrations during MR14-06 Leg 1 and backward trajectories (120 h). Magenta lines indicate cases in which trajectory entered regions over land (at altitudes < 2500 m a.s.l.). Red lines indicate cases in which observed O$_3$ concentrations exceeded 50 ppb.




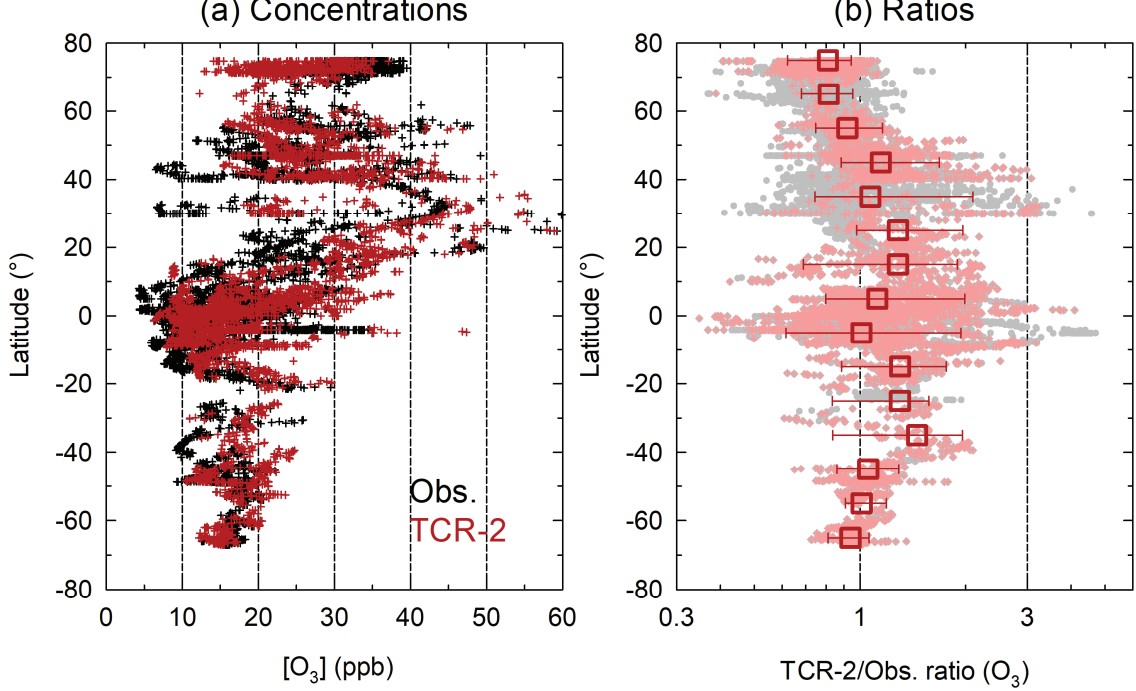

**Figure 4.** (a) Latitudinal distribution of $O_3$ concentrations from observation (black) and TCR-2 (red). Data are limited to cases of marine origins where difference between predicted and observed CO concentrations was less than 50 ppb ($N =$ 5647). (b) Latitudinal distribution of TCR-2/observation ratio for $O_3$ (light red dots) and binned medians (ranges are from 10th to 90th percentile) over 10°. Gray dots represent all cases without data selection.





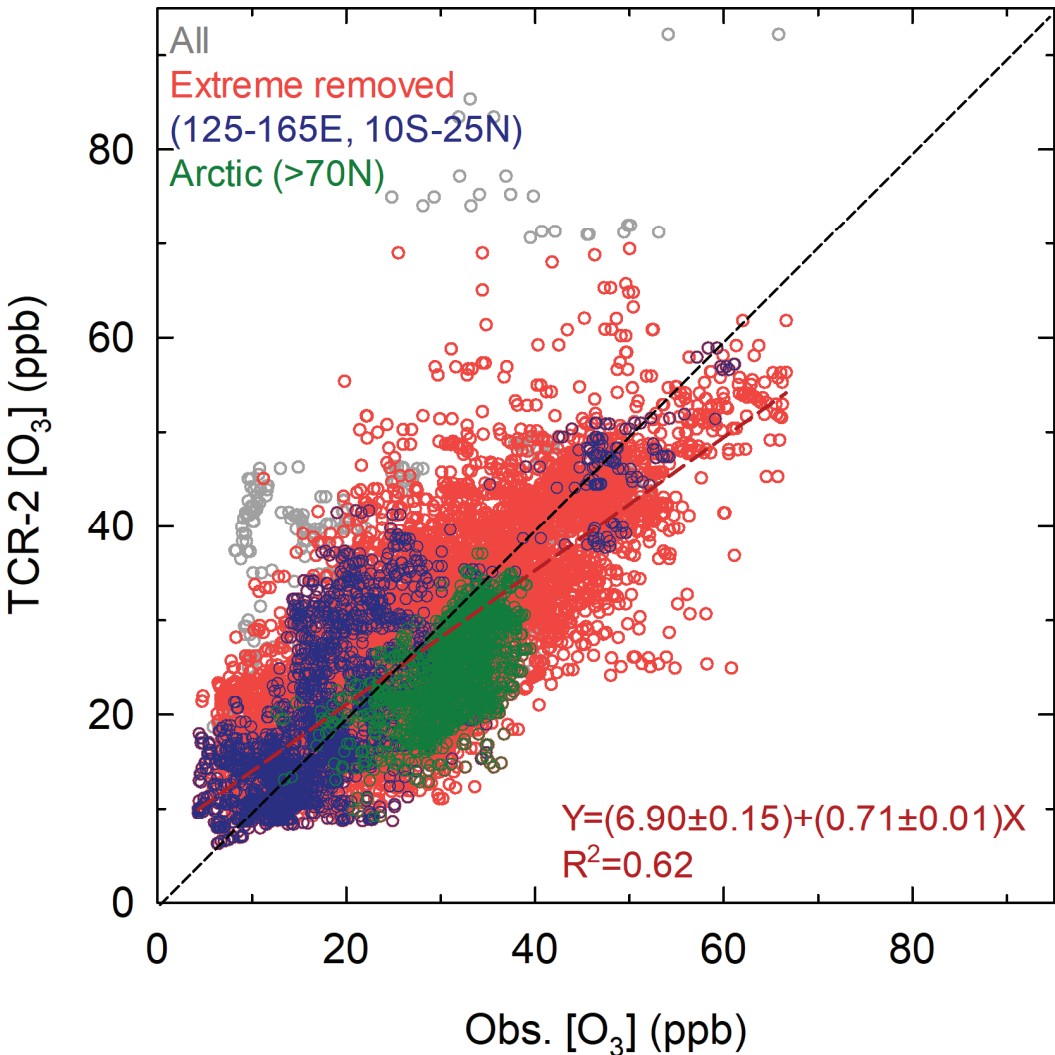

**Figure 5.** Scatterplot of observed and reanalysis $O_3$ concentrations. Gray circles include all data ($N = 11\ 666$), red circles are cases when data from MR14-01 and extreme cases when TCR-2 exceeded 70 ppb were removed. Green circles indicate data from Arctic region (>70° N) and blue circles indicate data from domain 2 (125–165° E, 10° S to 25° N).





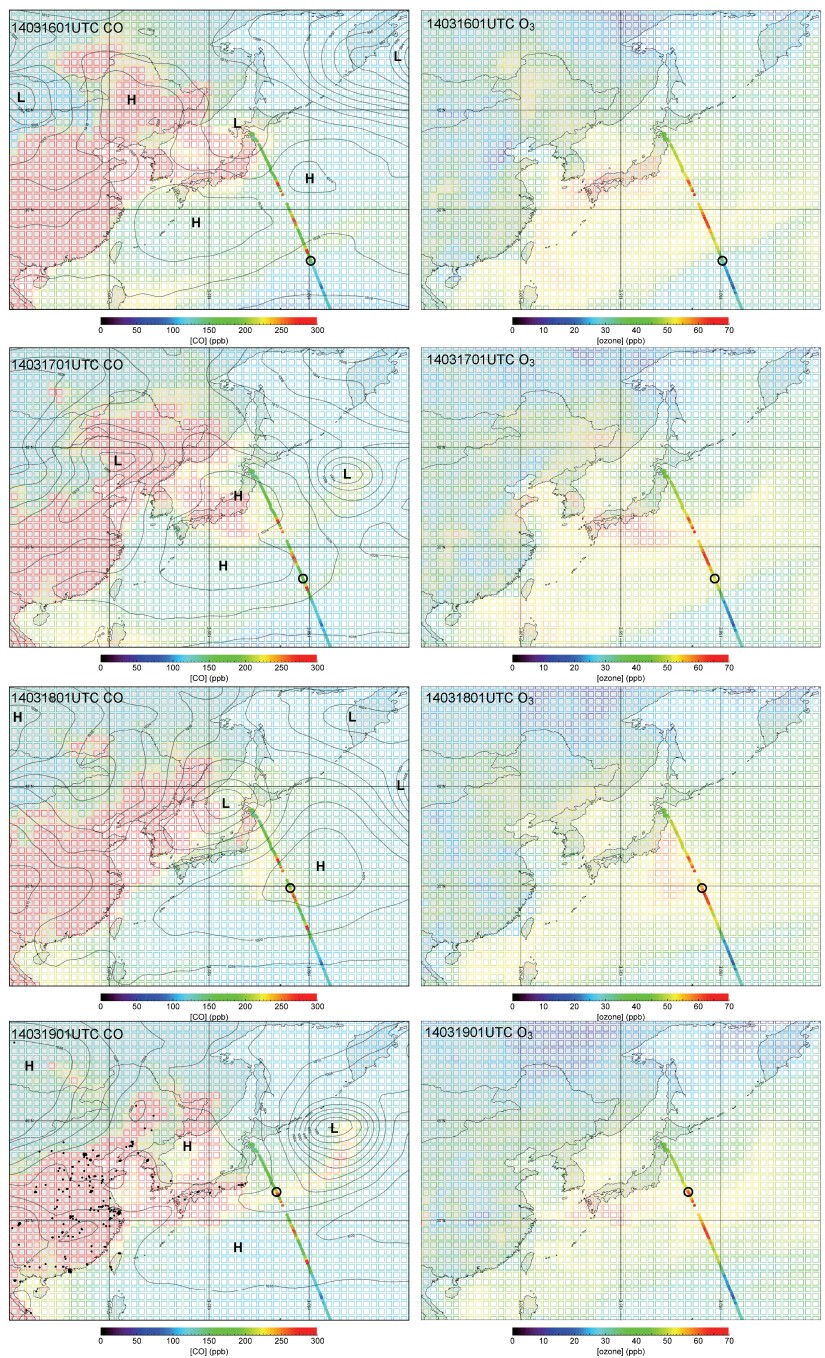

**Figure 6.** Temporal evolution of surface O$_3$ distribution from TCR-2 (colored open squares) and from observations on R/V *Mirai* (all data, position of vessel is black circle) during MR14-02. Black circles in 14031901UTC CO panel indicate locations with high anthropogenic CO emissions (EDGAR).



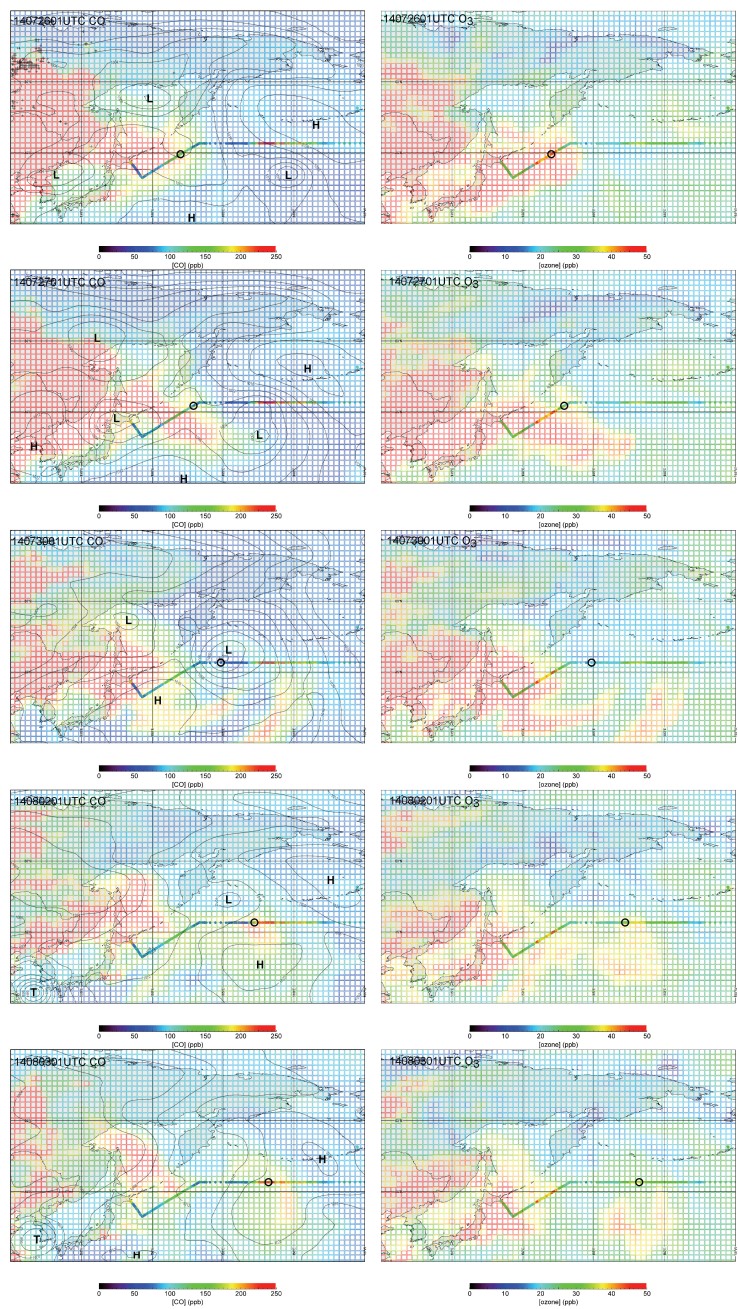

**Figure 7.** Temporal evolution of surface $O_3$ distribution from TCR-2 (colored open squares) and from observations on R/V *Mirai* (all data, position of vessel is shown by black circle) during MR14-04 Leg 2. Plus signs in top-left panel show points with high carbon emissions from wildfires (GFED).




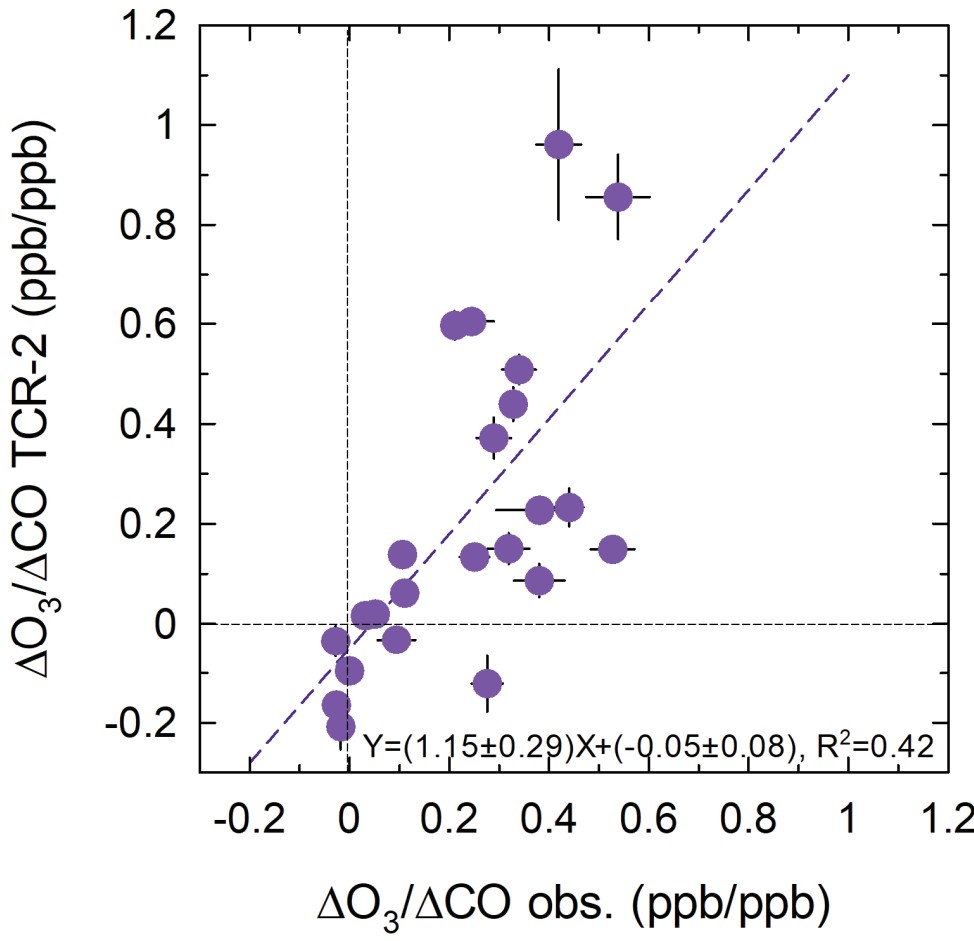

**Figure 8.** Correlations between $\Delta O_3/\Delta CO$ ratios from observations and TCR-2.




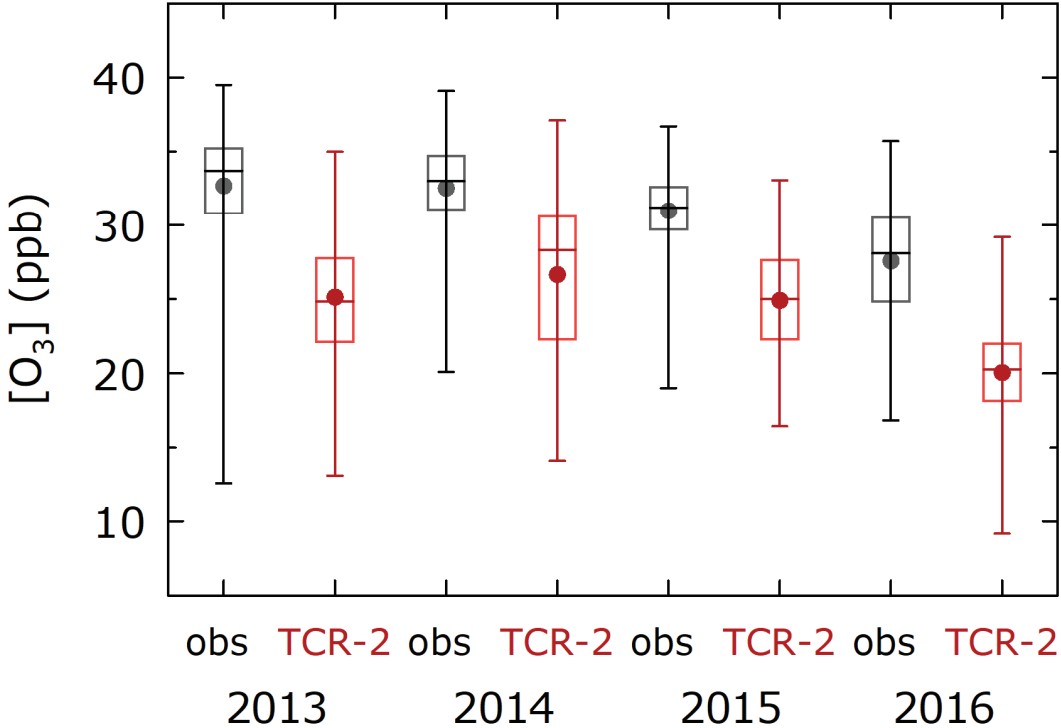

**Figure 9.** Repeated underestimations of O$_3$ concentration ranges by TCR-2 relative to observed values in Arctic region (>70° N). Boxes and horizontal bars indicate 75 %, 50 % (median), and 25 %, and whiskers indicate 90 % and 10 %. Circles are averages.





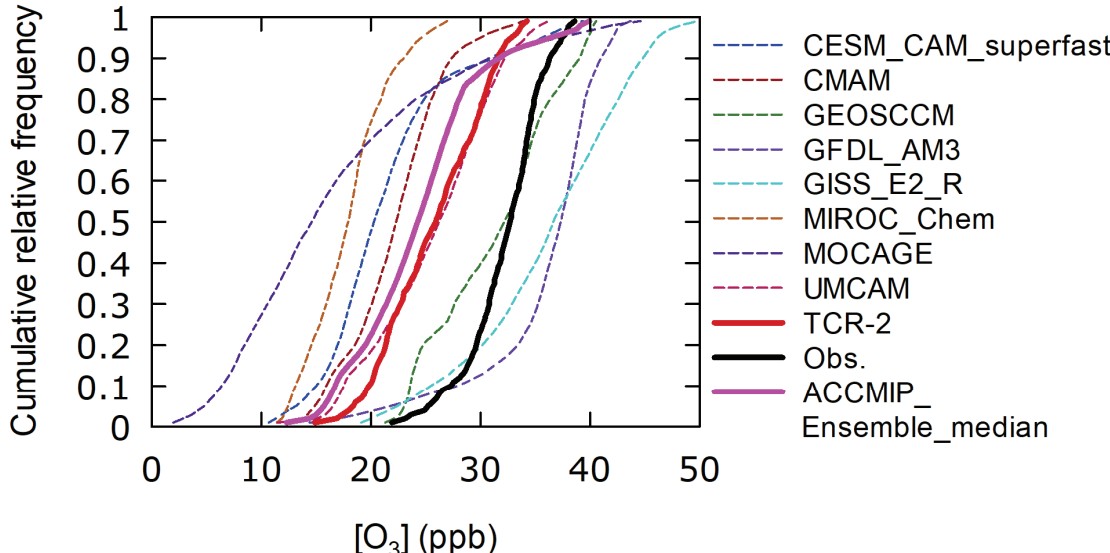

**Figure 10.** Cumulative relative frequency distributions of $O_3$ concentrations from observations (black), TCR-2 (red), and members and ensemble median of ACCMIP in Arctic grid 1 [72.5–77.5° N and 190–205° E (155–170° W)] in September.



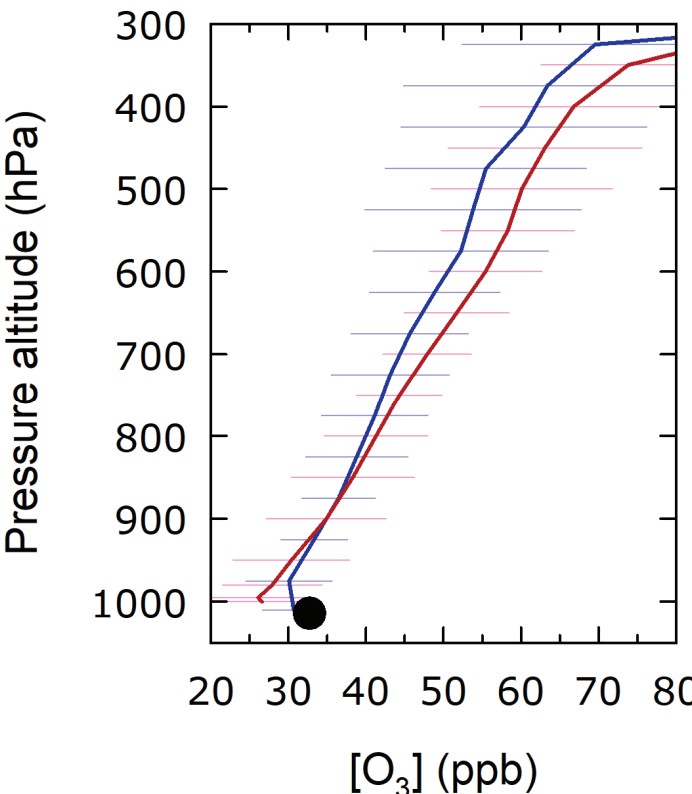

**Figure 11.** Average profile from $O_3$ soundings (blue) and that from TCR-2 (red). Average from surface observations on R/V *Mirai* is shown as black circle.



**Figure 12.** Cumulative relative frequency distributions of O₃ concentrations from observations (black), TCR-2 (red), and members and ensemble median of ACCMIP for domain 3 (0–15° N and 150–165° E) in (a) March and (b) December.





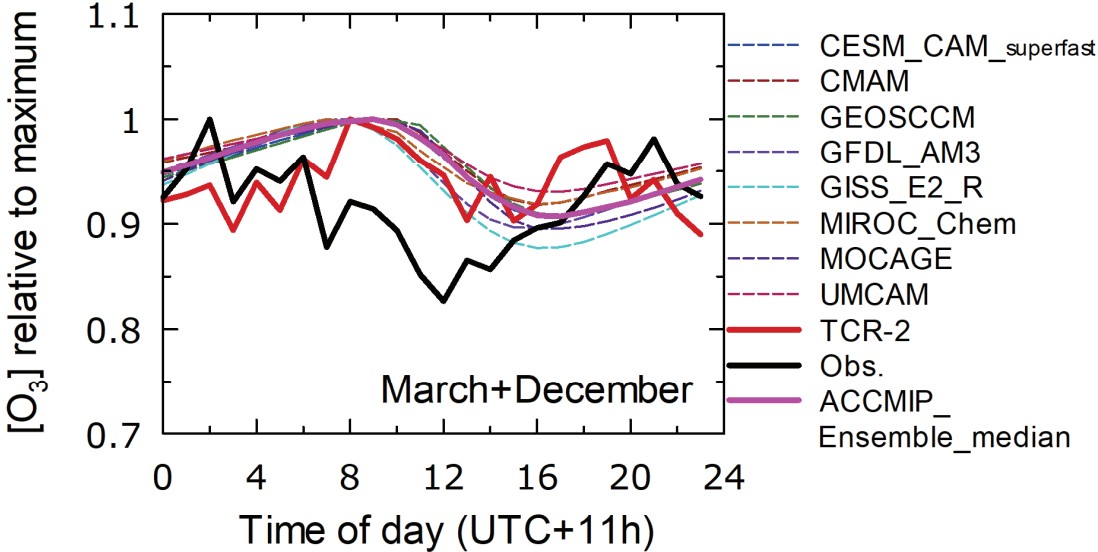

**Figure 13.** Diurnal variation patterns relative to maximum from observations (black), TCR-2 (red), and members and ensemble median of ACCMIP for domain 3 (0–15° N and 150–165° E). Data from March and December are merged.





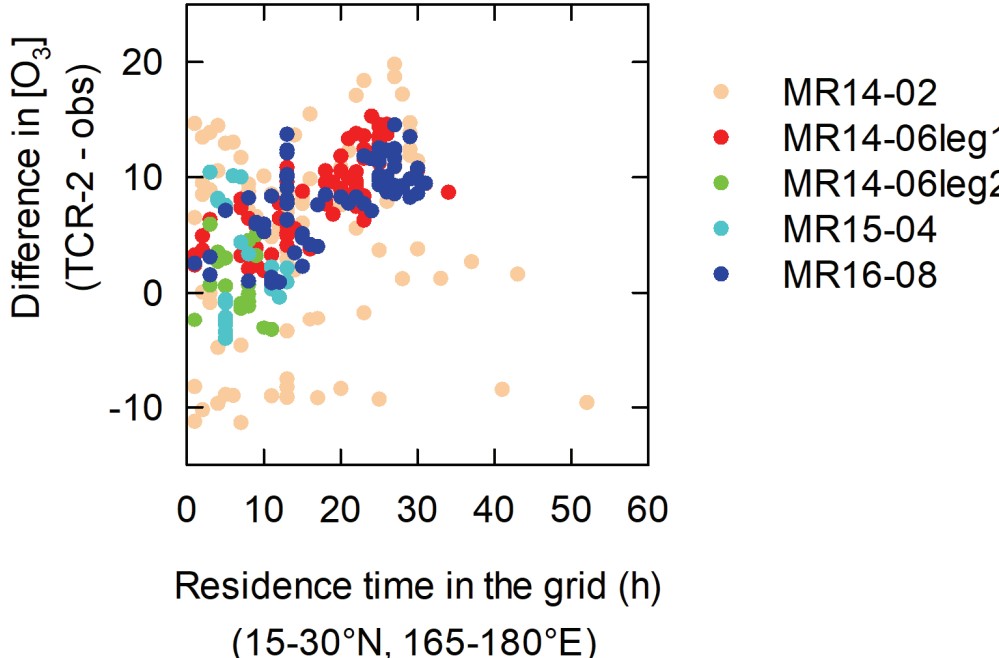

**Figure 14.** High biases in TCR-2 with respect to observations in domain 3 (0–15° N and 150–165° E) show positive correlations with daytime residence times in grid 15–30° N and 165–180° E.