# Peer review of "Ozone and carbon monoxide observations over open oceans on R/V *Mirai* from 67° S to 75° N during 2012 to 2017: Testing global chemical reanalysis in terms of Arctic processes, low ozone levels at low latitudes, and pollution transport"

_Atmospheric Chemistry and Physics, 2018_

## Referee Comment (RC1) · Anonymous Referee #1 · 9 Feb 2019

This manuscript provides a very nice and well-written presentation of a valuable data set of marine boundary layer ozone and CO observations, spanning the full latitudinal range of the North and South Pacific Oceans. As demonstrated in the paper, the data are valuable for understanding pollutant outflow from Asia and ozone destruction in the tropics. The data are also very useful for global atmospheric chemistry model evaluation. I recommend that the paper be published after a minor revision, as described below.

P1, line 21 Here the authors use the term "chemical transport models" as a general term to refer to all types of models that quantify atmospheric chemistry processes. As explained in the recent TOAR paper by Young et al. [2018], the general term should be "global atmospheric chemistry models".

P5, line 9 Here it says that the TCR-2 chemical reanalysis relies on assimilation of TES ozone values. TES provided relatively dense global coverage from 2004 to 2010, but after 2010 the instrument slowly lost power and its observational range was steadily reduced from global coverage to just a few urban areas. How did this reduction in coverage affect the TCR-2 ozone values?

When reviewing previous studies of ocean surveys, the following paper should be referenced. This early study reported increasing ozone across the Atlantic Ocean, using ship-borne observations: Lelieveld, J., Van Aardenne, J., Fischer, H., De Reus, M., Williams, J. and Winkler, P., 2004. Increasing ozone over the Atlantic Ocean. Science, 304(5676), pp.1483-1487.

In addition to the TOAR paper by Schultz et al. [2017], reference should also be made to Gaudel et al. [2018], as this is the TOAR paper that describes ozone observations at remote locations in order to understand the trends of ozone that are important for climate studies.

Gaudel, A., et al. (2018), Tropospheric Ozone Assessment Report: Present-day distribution and trends of tropospheric ozone relevant to climate and global atmospheric chemistry model evaluation. Elem Sci Anth, 6(1), p.39. DOI: http://doi.org/10.1525/elementa.291

P2, line 11 The radiative forcing of ozone needs to be stated with its uncertainty: 0.4 +/- 0.2 W m$-2$

Figure 1 It's difficult to see the magenta dots that indicate the TOAR observations. Please make the dots a little larger. Also panel (a) is presented below panel (b). This is confusing and the order should be reversed.

Figure 2 The color scale runs from 0 to 600 which leaves most data points in the blue range of colors. This provides very little contrast and makes it difficult to see concentration gradients. Please try lowering the maximum value on the color scale from 600 to 400 or 300. This should provide greater contrast.

Figure 3 It's difficult to distinguish between the red and magenta trajectories. Please try using different colors.

Throughout the paper there are many instances in which an ozone or CO mixing ratio is described as a concentration. Technically, this is not correct as a concentration has units of mass/volume. To be consistent with SI metrology, any value in units of ppbv needs to be described as a mixing ratio. Furthermore, ppb needs to be listed as ppbv.

---

## Referee Comment (RC2) · Anonymous Referee #2 · 14 Feb 2019

This paper discusses comprehensive shipborne O3 and CO measurements covering a large oceanic region from the Arctic to the Southern Ocean over the period of 2012-2017. The dataset was thoroughly analysed and was compared to the simulation results from a tropospheric chemistry reanalysis model (TCR-2), demonstrating the usefulness of such dataset in critical model evaluation. The authors also carried out two fo-

cused analyses assessing the underlying processes causing models to underestimate Arctic O3 and to overestimate O3 in the western Pacific equatorial region, respectively, compared to observations. The paper is very well written with detailed and in-depth analyses. The dataset is a significant addition to the current surface O3 database over remote oceanic regions and is valuable for model evaluations. I recommend the paper to be published after the authors have addressed some minor comments that are detailed below.

Specific comments:

Page 1, L31: "less efficient dry deposition" than assumed in the model sounds very speculative. Dry deposition coefficient is generally considered very slow over the ocean, and it is unlikely that there is much room for a significant impact when adjusting the dry deposition coefficient.

Page 1, L33: "the observed O3 level frequently decreased to ...": could add "more" before "frequently"

Page 3, L1-5: I am not sure CO2 observations are that relevant here.

Page 4, L27: please define "BC".

Page 5, 1st paragraph: Can you briefly describe the chemical mechanism used in the TCR-2 framework? This will inform a later discussion of photochemical production.

Page 6, L10 & 11: Suggest naming the regions where these locations are.

Page 6, L22 & 23: "South of ..." & "In equatorial regions..." seem overlap in what the authors try to convey; South of 15oN implies the equatorial region as well, if it doesn't go beyond 15oS.

Page 7, L29: Could you elaborate a bit more on "overestimate of photochemical O3 production"?

Page 9, L27, Is "less efficient production of O3" related to the chemical mechanism

used in the model?

Page 10, L20: Could you elaborate "Processes other than daytime photochemistry..."?

Page 11, L6: the significance can be established by Student's t test.

Page 11, L17: "unlike large increments obtained at low and..." – what does "large increments" refer to?

Page 11, L29-30: what is the dry deposition coefficient over this region used in the model? How does it compare with literature values for a similar land surface type? It is unlikely that dry deposition plays a significant role here, especially over the ocean; see above.

Page 11, L31-32: Can you put a reference here?

Page 11, L33: what is "(AMAP 2015)"?

Page 12, P14: It looks like the comparison between observations and ACCMIP models depends on the frequency ranges, and it is too general to claim the comparisons are poorer for December. Maybe you could elaborate on the seasonal difference in model performance? Is there any systematic model bias that are season dependent?

Page 12, L16-17: "The large variations among the model results could be the result of different assumptions regarding the dry deposition velocity of O3" – Do you have any reference to back this up? It surprises me that differences in dry deposition among the models over the ocean would result in such a large model spread. I'd rather think that differences amongst the models in the efficiency of transport of mid-latitude polluted air to the Arctic, coupled to maybe differences in mid-latitude ozone production, is likely the driving factor. The large model spread in the middle ranges in March might reflect the impact of large variations in transport.

Page14, L22: again, what is the dry deposition coefficient over the ocean used in here? See my previous comments regarding the unlikely impact from dry deposition

over the ocean. Many of the ACCMIP models use "off-line" dry deposition schemes characterized by prescribed, fixed dry deposition velocities over open water and ice which are documented in the literature. Hence an assertion that variations in these assumed dry deposition velocities drive the differences in the simulation of O3 needs to be backed up by a discussion of this literature and does not need to be the subject of speculation. Page14, L29: I would replace "later inter-comparisons" with "future intermodal comparisons".

Technical comments:

Page 6, L5 & L9: may replace "namely" with "i.e." Page 12, L5: Do you mean "studied region"? Figure 1: swap positions of (a) and (b), domain 3 is not clearly visible Figures 6 and 7: "CO" missing from the captions – should be "surface CO and O3" Figure 13: add "mixing ratios" after "maximum"

---

## Author Response (AR1)

**Response to the Reviewer #1**

This manuscript provides a very nice and well-written presentation of a valuable data set of marine boundary layer ozone and CO observations, spanning the full latitudinal range of the North and South Pacific Oceans. As demonstrated in the paper, the data are valuable for understanding pollutant outflow from Asia and ozone destruction in the tropics. The data are also very useful for global atmospheric chemistry model evaluation. I recommend that the paper be published after a minor revision, as described below.

We appreciate the reviewer's careful reading and positive comments on our manuscript. Detailed point-by-point responses are given below.

1) P1, line 21 Here the authors use the term "chemical transport models" as a general term to refer to all types of models that quantify atmospheric chemistry processes. As explained in the recent TOAR paper by Young et al. [2018], the general term should be "global atmospheric chemistry models".

The term "chemical transport models" is replaced with "global atmospheric chemistry models" to generally cover all types of model frameworks.

2) P5, line 9 Here it says that the TCR-2 chemical reanalysis relies on assimilation of TES ozone values. TES provided relatively dense global coverage from 2004 to 2010, but after 2010 the instrument slowly lost power and its observational range was steadily reduced from global coverage to just a few urban areas. How did this reduction in coverage affect the TCR-2 ozone values?

The following sentences will be added to clarify this point (page 5, lines 21-25):

"Because the number of assimilated TES  $O_3$  retrievals decreased substantially after 2010, the data assimilation performance became worse after 2010 in the previous version TCR-1 (Miyazaki et al., 2015), and it can be expected that TCR-2 has similar increases in  $O_3$  analysis errors after 2010. Nevertheless, the multi-constituent data assimilation framework provides comprehensive constraints on the chemical system and entire tropospheric  $O_3$  profiles through corrections made to precursors' emissions and stratospheric concentrations, as demonstrated by Miyazaki et al. (2015, 2019). "

3) When reviewing previous studies of ocean surveys, the following paper should be referenced. This early study reported increasing ozone across the Atlantic Ocean, using ship-borne observations: Lelieveld, J., Van Aardenne, J., Fischer, H., De Reus, M., Williams, J. and Winkler, P., 2004. Increasing ozone over the Atlantic Ocean. Science, 304(5676), pp.1483-1487.

In the revised manuscript, we will cite the paper as "a large collection from multiple cruises (e.g., Lelieveld

et al., 2004)" (page 2, line 31)

4) In addition to the TOAR paper by Schultz et al. [2017], reference should also be made to Gaudel et al. [2018], as this is the TOAR paper that describes ozone observations at remote locations in order to understand the trends of ozone that are important for climate studies.

Gaudel, A., et al. (2018), Tropospheric Ozone Assessment Report: Present day distribution and trends of tropospheric ozone relevant to climate and global atmospheric chemistry model evaluation. Elem Sci Anth, 6(1), p.39. DOI: http://doi.org/10.1525/elementa.291

We will mention Gaudel et al. (2018), together with Schultz et al. (2017), in the revised manuscript (page 2, line 22).

5) P2, line 11 The radiative forcing of ozone needs to be stated with its uncertainty:  $0.4 \pm 0.2$  W m-2

We will include the uncertainty range as suggested (page 2, line 13).

6) Figure 1 It's difficult to see the magenta dots that indicate the TOAR observations. Please make the dots a little larger. Also panel (a) is presented below panel (b). This is confusing and the order should be reversed.

The size of the magenta dots is increased. Panel (a) will be put on top in the revised manuscript.

7) Figure 2 The color scale runs from 0 to 600 which leaves most data points in the blue range of colors. This provides very little contrast and makes it difficult to see concentration gradients. Please try lowering the maximum value on the color scale from 600 to 400 or 300. This should provide greater contrast.

After revision, the scale will be from 40 to 340 ppbv for CO mixing ratio to have better contrast.

8) Figure 3 It's difficult to distinguish between the red and magenta trajectories. Please try using different colors.

The red thin lines in Figure 3 will be dark-red, thicker lines in the revised manuscript.

9) Throughout the paper there are many instances in which an ozone or CO mixing ratio is described as a concentration. Technically, this is not correct as a concentration has units of mass/volume. To be consistent with SI metrology, any value in units of ppbv needs to be described as a mixing ratio. Furthermore, ppb needs to be listed as ppbv.

As suggested, "mixing ratio" and ppbv are used, in the revised manuscript.

Finally, we added a co-author Takashi Sekiya, who contributed significantly to development of data assimilation system and TCR-2.

We thank the reviewer again for the productive comments.

**References**

- Gaudel, A., Cooper, O. R., Ancellet, G., Barret, B., Boynard, A., Burrows, J. P., Clerbaux, C., Coheur, P.-F., Cuesta, J., Cuevas, E., Doniki, S., Dufour, G., Ebojie, F., Foret, G., Garcia, O., Granados-Muñoz, M. J., Hannigan, J. W., Hase, F., Hassler, B., Huang, G., Hurtmans, D., Jaffe, D., Jones, N., Kalabokas, P., Kerridge, B., Kulawik, S., Latter, B., Leblanc, T., Le Flochmoën, E., Lin, W., Liu, J., Liu, X., Mahieu, E., McClure-Begley, A., Neu, J. L., Osman, M., Palm, M., Petetin, H., Petropavlovskikh, I., Querel, R., Rahpoe, N., Rozanov, A., Shultz, M. G., Schwab, J., Siddans, R., Smale, D., Steinbacher, M., Tanimoto, H., Tarasick, D. W., Thouret, V., Thompson, A. M., Trickl, T., Weatherhead, E., Wespes, C., Worden, H. M., Vigouroux, C., Xu, X., Zeng, G., and Ziemke, J.: Tropospheric Ozone Assessment Report: Present day distribution and trends of tropospheric ozone relevant to climate and global atmospheric chemistry model evaluation. Elem Sci Anth, 6(1), p.39. DOI: http://doi.org/10.1525/elementa.291, 2018.
- Lelieveld, J., Van Aardenne, J., Fischer, H., De Reus, M., Williams, J. and Winkler, P., 2004. Increasing ozone over the Atlantic Ocean. Science, 304(5676), 1483-1487.

**Response to the Reviewer #2:**

We thank the reviewer very much for reading our paper carefully and giving us valuable comments. Detailed responses to the comments are given below.

This paper discusses comprehensive shipborne O3 and CO measurements covering a large oceanic region from the Arctic to the Southern Ocean over the period of 2012-2017. The dataset was thoroughly analysed and was compared to the simulation results from a tropospheric chemistry reanalysis model (TCR-2), demonstrating the usefulness of such dataset in critical model evaluation. The authors also carried out two focused analyses assessing the underlying processes causing models to underestimate Arctic O3 and to overestimate O3 in the western Pacific equatorial region, respectively, compared to observations. The paper is very well written with detailed and in-depth analyses. The dataset is a significant addition to the current surface O3 database over remote oceanic regions and is valuable for model evaluations. I recommend the paper to be published after the authors have addressed some minor comments that are detailed below.

**Specific comments:**

 Page 1, L31: "less efficient dry deposition" than assumed in the model sounds very speculative. Dry deposition coefficient is generally considered very slow over the ocean, and it is unlikely that there is much room for a significant impact when adjusting the dry deposition coefficient.

The discussion on dry deposition in the Arctic is revised.

The dry deposition velocity of TCR-2 (CHASER) on the Arctic ocean surface, ~0.04 cm s-1, is on the high side of ~0.01–0.05 cm s-1, a range adopted into global atmospheric chemistry models. This is now clarified in the revised manuscript (first in page 5, lines 16-20 in the model description section and then in page 12, lines 7-9 in the discussion section on the Arctic processes) as follows:

"The dry deposition velocity  $(v_d)$  of O3 is computed as  $(r_a + r_b + r_s)^{-1}$ , where  $r_a$ ,  $r_b$ , and  $r_s$  are the aerodynamic resistance, the surface canopy (quasi-laminar) layer resistance, and the surface resistance, respectively (Wesely, 1989).  $1/r_s$  over ocean surface was assumed to be 0.075 cm s-1 globally, irrespective of regions (Sudo et al., 2002). As a result,  $v_d$  was ~0.04 cm s-1 over the Arctic open ocean in September, for instance. This will be a subject of discussion in Sect. 3.3.2."

"The  $v_d$ , ~0.04 cm s-1 over the Arctic open ocean in September for CHASER (TCR-2), is on the high side of ~0.01–0.05 cm s-1, a range adopted into global atmospheric chemistry models (see Fig. 4 of Hardacre et al., 2015)."

Then we cited Ganzeveld et al. (2009), having discussed potential impact of  $v_d$  change on the surface ozone concentrations in high-latitude regions (page 12, lines 9-10):

"Ganzeveld et al. (2009) discussed a sensitivity study shifting their standard  $v_d$  of 0.05 cm s-1 to 0.01 cm s-1 substantially increased surface ozone concentrations by up to 60% in high-latitude regions."

Indeed, with  $v_d = 0.04$  cm s-1, assuming 500 m of boundary layer height, the lifetime of ozone due to dry deposition would be as low as about 14 days in a rough calculation. As the photochemical loss over the Arctic is weak, dry deposition can be an important loss process. In the future we will perform a sensitivity test with different assumption with  $v_d$

On the basis of this discussion, in Abstract, the previous sentence is kept but is now clarified that the sentence is for CHASER (TCR-2) (page 1, line 33): "For TCR-2 (CHASER), dry deposition on the Arctic ocean surface might also have been overestimated."

2) Page 1, L33: "the observed O3 level frequently decreased to . . .": could add "more" before "frequently"

The sentence is revised accordingly.

3) Page 3, L1-5: I am not sure CO2 observations are that relevant here.

We agree that  $CO_2$  is not that relevant to the main topic of this manuscript, i.e., analysis of reactive species. However, we still think it important to learn the strategy how the network observations are maintained and how the data are widely collected. Thus, the previous sentences are preserved.

4) Page 4, L27: please define "BC".

We mention "black carbon" in the revised manuscript.

5) Page 5, 1st paragraph: Can you briefly describe the chemical mechanism used in the TCR-2 framework? This will inform a later discussion of photochemical production.

In the revised manuscript, we will include the following sentences (page 5, lines 14-16):

"The base forward model CHASER V4.0 used for TCR-2 has been described by Sekiya et al. (2018). Briefly, 93 species and 263 reactions (including heterogeneous reactions) represent  $O_x$ -NOx-HOx-CH4-CO photochemistry and oxidation of non-methane volatile organic compounds. Tropospheric halogen chemistry is not included."

6) Page 6, L10 & 11: Suggest naming the regions where these locations are.

We mention " the western Pacific equatorial region" in the revised manuscript.

7) Page 6, L22 & 23: "South of . . ." & "In equatorial regions. . ." seem overlap in what the authors try to convey; South of 150N implies the equatorial region as well, if it doesn't go beyond 150S.

This part is revised as follows in the revised manuscript (page 7, lines 2-3):

"South of  $15^{\circ}$  N, even lower levels were dominant, i.e., <15 ppbv; particularly in equatorial regions, levels less than 10 ppbv of O3 were frequently observed."

8) Page 7, L29: Could you elaborate a bit more on "overestimate of photochemical O3 production"?

In the revised manuscript we will mention overestimation of photochemical O3 production during long-range transport in TCR-2 (page 8, line 7).

9) Page 9, L27, Is "less efficient production of O3" related to the chemical mechanism used in the model?

This part is describing a single case. Considering possibilities that J values and other important parameters are also not well represented in the model, it is difficult to relate the less efficient production of  $O_3$  specifically to the chemical mechanism.

**10) Page 10, L20: Could you elaborate "Processes other than daytime photochemistry. . . "?**

We will mention "processes other than daytime photochemistry (e.g., nocturnal chemistry)" after revision.

11) Page 11, L6: the significance can be established by Student's t test.

Welch's t test, allowing unequal variances in the two group, was made. In the revised manuscript, we will mention that "the reanalysis significantly underestimated this (24.6 ppbv) based on Welch's *t* test (p < 0.001)."

12) Page 11, L17: "unlike large increments obtained at low and. . ." – what does "large increments" refer to?

The sentence will be rewritten as follows:

"The reanalysis ozone over the Arctic Ocean can be similar to the model predictions, except when poleward transports are strong enough to propagate observational information from low and mid latitudes."

13) Page 11, L29-30: what is the dry deposition coefficient over this region used in the model? How does it compare with literature values for a similar land surface type? It is unlikely that dry deposition plays a significant role here, especially over the ocean; see above.

For the Arctic, the assumed dry deposition velocity over ocean was ~0.04 cm s-1 for CHASER (TCR-2), at the high side of a range often used in global atmospheric chemistry models (0.01-0.05 cm s-1). A rough calculation assuming 500 m of boundary layer height, the lifetime of ozone due to dry deposition over ocean is calculated to be as 14 days. For the air masses traveling over ocean for >3 days, which were typical for observations, dry deposition may have potential to reduce O3 levels by ~20%. Over the Arctic, as the photochemical loss is weak, dry deposition can be an important loss process. Indeed, Ganzeveld et al. (2009) discussed a sensitivity study shifting their standard  $v_d$  of 0.05 cm s-1 to 0.01 cm s-1 substantially increased surface ozone concentrations by up to 60% in high-latitude regions (page 12, lines 9-10). In the future we will perform a sensitivity test with different assumption with  $v_d$ .

**14) Page 11, L31-32: Can you put a reference here?**

McClure-Begley et al., 2014 is added.

**15) Page 11, L33: what is "(AMAP 2015)"?**

It is a report from AMAP, Arctic Monitoring and Assessment Programme, Arctic Council, as shown in the reference list. Now the reference is given correctly as (AMAP, 2015).

16) Page 12, P14: It looks like the comparison between observations and ACCMIP models depends on the frequency ranges, and it is too general to claim the comparisons are poorer for December. Maybe you could elaborate on the seasonal difference in model performance? Is there any systematic model bias that are season dependent?

As found in the subsection title, our focus here is to compare occurrence of < 10 ppbv in the model and observational datasets. And as the observations are basically limited to the two months, it is difficult to discuss the seasonal difference in detail. Thus we will just simply add " for any of the percentiles" to explain the poor comparison for December in the revised manuscript (page 12, lines 28-30):

"For December (Fig. 12b), the performances of the models were poorer; although CESM-CAM-super fast and GISS-E2-R again captured the observed distributions in low ranges, all others, including the ACCMIP ensemble median and TCR-2, overestimated the mixing ratios for any of the percentiles."

17) Page 12, L16-17: "The large variations among the model results could be the result of different assumptions regarding the dry deposition velocity of O3" – Do you have any reference to back this up? It surprises me that differences in dry deposition among the models over the ocean would result in such a large model spread. I'd rather think that differences amongst the models in the efficiency of transport of mid-latitude polluted air to the Arctic, coupled to maybe differences in mid-latitude ozone production, is

likely the driving factor. The large model spread in the middle ranges in March might reflect the impact of large variations in transport.

For this part, discussing low  $O_3$  levels at low latitudes, we will delete a previous speculative sentence regarding possible variability in the dry deposition velocity among the ACCMIP models in the revised manuscript, as the reviewer pointed out. Instead, we added that "the large variations among the model results may reflect the impact of large variations in transport, particularly in March", as suggested by the reviewer (from page 12, lines 30-31).

18) Page14, L22: again, what is the dry deposition coefficient over the ocean used in here? See my previous comments regarding the unlikely impact from dry deposition over the ocean. Many of the ACCMIP models use "off-line" dry deposition schemes characterized by prescribed, fixed dry deposition velocities over open water and ice which are documented in the literature. Hence an assertion that variations in these assumed dry deposition velocities drive the differences in the simulation of O3 needs to be backed up by a discussion of this literature and does not need to be the subject of speculation.

As discussed above, we still think that the relatively fast dry deposition assumed in the CHASER (TCR-2) would at least partially explain the lower-than-observed  $O_3$  levels over the Arctic. The sentence in Sect. 4 is retained but is now clarified that this sentence is for TCR-2.

19) Page14, L29: I would replace "later inter-comparisons" with "future intermodal comparisons".

We will revise accordingly.

**20) Technical comments:**

Page 6, L5 & L9: may replace "namely" with "i.e." Page 12, L5: Do you mean "studied region"? Figure 1: swap positions of (a) and (b), domain 3 is not clearly visible Figures 6 and 7: "CO" missing from the captions – should be "surface CO and O3" Figure 13: add "mixing ratios" after "maximum"

We will revise accordingly.

Finally, we added a co-author Takashi Sekiya, who contributed significantly to development of data assimilation system and TCR-2.

We again thank the reviewer for the important suggestions.

**Data availability**

The observational data set for O3 and CO obtained on R/V Mirai for the study period (2012-2017) is collectively available

25 from https://ebcrpa.jamstec.go.jp/atmoscomp/obsdata/ as a single text file and from http://www.godac.jamstec.go.jp/darwin/e for individual cruise legs. TCR-2 reanalysis data are available from http://ebcrpa.jamstec.go.jp/tcr2/download.html.

**Author contribution**

YK (Kanaya) designed the study, conducted analyses, wrote the manuscript, managed the instruments and created the observational data set. KM, TS and KS (Sudo) created TCR-2 data. FT, TM, HT, YK (Komazaki), XP and SK substantially
 contributed to shipborne observations and preparation. HT and SK also provided supporting data from MAX-DOAS

30

observations. JI, KS (Sato) and KO conducted shipborne ozone soundings and provided the data. All co-authors provided professional comments to improve the manuscript.

**Competing interest**

The authors declare that they have no conflict of interest.

5

**Acknowledgments**

We gratefully acknowledge assistance from the Principal Investigators of all cruises and support from Global Ocean Development Inc. and Nippon Marine Enterprise, Ltd. This research was supported by the Coordination Funds for Promoting AeroSpace Utilization and by the Arctic Challenge for Sustainability (ArCS) project of the Ministry of Education,

- 10 Culture, Sports, Science and Technology (MEXT), MEXT/JSPS KAKENHI Grant Numbers 24241009, 18H04143 and 18H01285, and the Environment Research and Technology Development Fund (2-1803) of the Ministry of the Environment, Japan. We thank Irina Petropavlovskikh (NOAA/ESRL) for providing data for Barrow. Part of the research was carried out at the Jet Propulsion Laboratory, California Institute of Technology, under a contract with the National Aeronautics and Space Administration. We thank Helen McPherson, PhD, from Edanz Group (www.edanzediting.com/ac) for editing a draft
- 15 of this manuscript.

[revised manuscript text omitted]
                            | 00:00UTC               | Mutsu         | 08:00UTC                | Onahama       |                                                                |
|               | Pacific                                  | 4 Jun 2012             |               | 24 Jun 2012             | (out of port) |                                                                |
| MR12-02 Leg2  | Western North                            | 08:00UTC               | Onahama       | 00:00UTC                | Mutsu         | Hachinohe Port from                                            |
|               | Pacific                                  | 24 Jun 2012            | (out of port) | 12 Jul 2012             |               | 0030 to 0800 UTC Jul
11 (no data)                           |
| MR13-04       | Western North                            | 23:50UTC               | Yokohama      | 00:00UTC                | Mutsu         |                                                                |
|               | Pacific                                  | 9 Jul 2013             |               | 29 Jul 2013             |               |                                                                |
| MR13-05       | Bering Sea                               | 23:50UTC               | Mutsu         | 17:40UTC                | Dutch Harbor  |                                                                |
|               | -                                        | 12 Aug 2013            |               | 26 Aug 2013             |               |                                                                |
| MR13-06 Leg1  | Arctic Ocean,                            | 18:00UTC               | Dutch Harbor  | 18:40UTC                | Dutch Harbor  |                                                                |
| _             | Bering Sea                               | 28 Aug 2013            |               | 7 Oct 2013              |               |                                                                |
| MR13-06 Leg2  | Bering Sea,                              | 17:40UTC               | Dutch Harbor  | 23:50UTC                | Mutsu         |                                                                |
|               | North Pacific                            | 9 Oct 2013             |               | 20 Oct 2013             |               |                                                                |
| MR14-01       | East Indian                              | 23:00UTC               | Mutsu         | 00:00UTC                | Palau         |                                                                |
|               | Ocean,
Equatorial
Region           | 8 Jan 2014             |               | 13 Feb 2014             |               |                                                                |
| MR14-02       | Western Pacific.                         | 00:00UTC               | Koror, Palau  | 00:00UTC                | Mutsu         | Hachinohe Port.                                                |
|               | Equatorial
Region                     | 15 Feb 2014            |               | 23 Mar 2014             |               | 0400-0900UTC 21
Mar 2014 (no data)                          |
| MR14-04 Leg1  | Western North                            | 22:10UTC               | Yokohama      | 04:00UTC                | Kushiro       |                                                                |
|               | Pacific                                  | 8 Jul 2014             |               | 15 Jul 2014             |               |                                                                |
| MR14-04 Leg2  | North Pacific                            | 01:00UTC               | Kushiro       | 17:50UTC                | Dutch Harbor  |                                                                |
|               |                                          | 17 Jul 2014            |               | 29 Aug 2014             |               |                                                                |
| MR14-05       | Arctic Ocean,                            | 18:10UTC               | Dutch Harbor  | 00:20UTC                | Yokohama      |                                                                |
|               | Bering Sea,
North Pacific             | 31 Aug 2014            |               | 10 Oct 2014             |               |                                                                |
| MR14-06 Leg1  | Western Pacific,
Equatorial
region | 06:10UTC
4 Nov 2014 | Mutsu         | 23:20UTC
17 Dec 2014 | Chuuk         | Yokohama Port, from
2310UTC 5 Nov to
0700UTC 7 Nov (with |
|               | 5                                        |                        |               |                         |               | data)                                                          |
| MR14-06 Leg2  | Western Pacific                          | 00:07UTC               | Chuuk         | 00:10UTC                | Palau         | · · · · · · · · · · · · · · · · · · ·                          |
|               | Equatorial region                        | 20 Dec 2014            |               | 19 Jan 2015             |               |                                                                |
| MR14-06 Leg3  | Western Pacific,                         | 00:00UTC               | Palau         | 00:00UTC                | Mutsu         | Hachinohe Port, from                                           |
|               | East Indian                              | 22 Jan 2015            |               | 25 Feb 2015             |               | 2330UTC 23 Feb to                                              |
|               | Ocean                                    |                        |               |                         |               | 0700UTC 24 Feb                                                 |
|               | Equatorial                               |                        |               |                         |               | (with data)                                                    |
|               | Region                                   |                        |               |                         |               |                                                                |
| MR15-03 leg 1 | North Pacific,                           | 22:50UTC               | Mutsu         | 18:50UTC                | Dutch Harbor  |                                                                |
|               | Bering Sea,                              | 23 Aug 2015            |               | 6 Oct 2015              |               |                                                                |
| MD45 00 Lun 0 | Arctic Ocean                             |                        | Dutch Hawken  |                         | N.A. A        | Hashingha Dart from                                            |
| MR15-03 leg 2 | Bering Sea,                              | 18:1001C               | Dutch Harbor  | 23:5001C                | Mutsu         | Hachinohe Port, from                                           |
|               | North Pacific                            | 9 OCI 2015             |               | 21 OCI 2015             |               | 230001C 20 OCI 10                                              |
|               |                                          |                        |               |                         |               | (with data)                                                    |
| MR15-04       | Western Pacific                          |                        | Muteu         | 02.2011TC               | lakarta       | Hachinghe Port from                                            |
| 1011110-04    | East Indian                              | 5 Nov 2015             | Matsa         | 20 Dec 2015             | Janarta       | 2250UTC 5 Nov to                                               |
|               | Ocean                                    |                        |               |                         |               | 0620UTC 6 Nov (with                                            |
|               | Equatorial                               |                        |               |                         |               | data)                                                          |
|               | Region                                   |                        |               |                         |               | , ·                                                            |
| MR15-05 leg 1 | East Indian                              | 03:10UTC               | Jakarta       | 00:50UTC                | Bali          |                                                                |
|               | Ocean                                    | 23 Dec 2015            |               | 11 Jan 2016             |               |                                                                |
| MR15-05 lea 2 | East Indian                              | 01:00UTC               | Benoa, Bali   | 23:50UTC                | Yokohama      |                                                                |

|               | Ocean, Western
North Pacific               | 13 Jan 2016             |              | 24 Jan 2016             |             |                                                                                                                               |
|---------------|-----------------------------------------------|-------------------------|--------------|-------------------------|-------------|-------------------------------------------------------------------------------------------------------------------------------|
| MR16-06       | Arctic Ocean,
Bering Sea,
North Pacific | 00:00UTC
22 Aug 2016 | Hachinohe    | 00:00UTC
5 Oct 2016  | Mutsu       | Nome port, 1600-
2010UTC 23 Sep
(with data);
Hachinohe Port from
2230UTC 3 Oct to
0720UTC 4 Oct (with
data) |
| MR16-08       | Western Pacific
Equatorial
Region       | 07:00UTC
27 Nov 2016 | Shimizu      | 21:00UTC
23 Dec 2016 | Suva        |                                                                                                                               |
| MR16-09 leg 1 | South Pacific                                 | 17:10UTC
26 Dec 2016 | Suva         | 11:00UTC
17 Jan 2017 | Puerto Mont |                                                                                                                               |
| MR16-09 leg 3 | Southern Ocean                                | 13:10UTC
8 Feb 2017  | Punta Arenas | 21:00UTC
4 Mar 2017  | Auckland    |                                                                                                                               |
| MR16-09 leg 4 | Western Pacific                               | 21:20UTC
7 Mar 2017  | Auckland     | 00:00UTC
28 Mar 2017 | Mutsu       | Hachinohe Port, from
2240UTC 26 Mar to
0650UTC 27 Mar
(with data)                                                    |

| PP-1 |             | _)•         |                 |         |       |                          |         |                          |       |         |       |         |
|------|-------------|-------------|-----------------|---------|-------|--------------------------|---------|--------------------------|-------|---------|-------|---------|
| cas  | Start time  | End time    | Latitude        | Longitu | R     | $\Delta O_3 / \Delta CO$ | R       | $\Delta O_3 / \Delta CO$ | COmax | COmax   | O₃max | O₃max   |
| е    | Start time  |             | (°N)            | de (°E) | (obs) | (obs)                    | (TCR-2) | (TCR-2)                  | (obs) | (TCR-2) | (obs) | (TCR-2) |
| А    | 22:00UTC    | 22:00UTC    | 41.32–          | 151.81– | 0.81  | 0.42±0.04                | 0.68    | 0.96±0.15                | 101.6 | 87.5    | 38.2  | 40.0    |
|      | 15 Aug 2013 | 17 Aug 2013 | 45.22           | 162.71  |       |                          |         |                          |       |         |       |         |
| В    | 12:00UTC    | 12:00UTC    | 54.47-          | 182.07- | 0.79  | 0.28±0.03                | -0.30   | -0.12±0.06               | 101.9 | 102.7   | 33.1  | 26.0    |
|      | 23 Aug 2013 | 25 Aug 2013 | 56.59           | 193.25  |       |                          |         |                          |       |         | l     |         |
| С    | 00:00 010   | 22:00 010   | 56.95-          | 191.36- | 0.80  | 0.54±0.06                | 0.85    | 0.86±0.08                | 116.4 | 117.2   | 43.1  | 38.7    |
|      | 05 UCI 2013 | 06 UCI 2013 | 40.09           | 192.82  |       |                          |         |                          |       |         |       |         |
| D    | 23.00 01C   | 23.00 01C   | 40.00-          | 143.72- | 0.75  | 0.32±0.04                | 0.58    | 0.15±0.03                | 137.4 | 129.2   | 51.1  | 40.9    |
|      | 15:00 LITC  | 15:00 LITC  | 25 21_          | 126.85  |       |                          |         |                          |       |         |       |         |
| E    | 12.lan 2014 | 14 Jan 2014 | 31.33           | 136.28  | 0.66  | 0.03±0.01                | 0.29    | 0.01±0.01                | 331.2 | 269.7   | 52.8  | 45.8    |
|      | 00:00 UTC   | 00:00 UTC   | 11 51-          | 151 64- |       |                          |         |                          |       |         |       |         |
| F    | 13 Mar 2014 | 15 Mar 2014 | 18 95           | 154 58  | 0.83  | 0.34±0.03                | 0.93    | 0.51±0.03                | 137.4 | 125.3   | 35.4  | 37.3    |
| _    | 00:00 UTC   | 00:00 UTC   | 25.08-          | 145.21- |       |                          |         |                          |       |         |       |         |
| G    | 17 Mar 2014 | 19 Mar 2014 | 34.12           | 149.15  | 0.86  | 0.11±0.01                | 0.75    | 0.06±0.01                | 299.1 | 219.0   | 66.6  | 56.3    |
|      | 13:00 UTC   | 10:00 UTC   | 42.67-          | 152.68- | 0.57  | 0.00.00.00               | 0.00    | 0.00.00.01               | 400.0 | 000.0   | 10.0  | 47.0    |
| н    | 24 Jul 2014 | 26 Jul 2014 | 45.32           | 157.02  | 0.57  | 0.38±0.09                | 0.92    | 0.23±0.01                | 132.2 | 208.0   | 48.2  | 47.9    |
| -    | 00:00 UTC   | 00:00 UTC   | 46.99–          | 171.62- | 0.00  | 0 11+0 01                | 0.03    | 0 14+0 01                | 240.4 | 220.0   | 38.0  | 37.2    |
|      | 02 Aug 2014 | 04 Aug 2014 | 47.01           | 177.97  | 0.90  | 0.1110.01                | 0.95    | 0.14±0.01                | 249.4 | 220.0   | 30.0  | 57.2    |
|      | 00:00 UTC   | 00:00 UTC   | 51.15–          | 208.00- | 0.87  | 0 53+0 04                | 0 70    | 0 15+0 02                | 110 0 | 116 1   | 35.4  | 31.5    |
| 5    | 25 Aug 2014 | 27 Aug 2014 | 53.38           | 221.14  | 0.07  | 0.03±0.04                | 0.70    | 0.15±0.02                | 113.0 | 110.1   | 55.4  | 51.5    |
| к    | 00:00 UTC   | 00:00 UTC   | 26.61-          | 141.81– | 0.92  | 0 44+0 03                | 0.67    | 0 23+0 04                | 154 4 | 145 1   | 55.6  | 46.4    |
|      | 08 Nov 2014 | 10 Nov 2014 | 32.60           | 149.90  | 0.02  | 0.1120.00                | 0.07    | 0.2020.01                | 101.1 | 110.1   | 00.0  | 10.1    |
| 1    | 02:00 UTC   | 21:00 UTC   | -1.60–          | 89.95-  | 0.95  | 0 21+0 01                | 0.96    | 0 60+0 03                | 181 1 | 105 7   | 31.8  | 24 4    |
|      | 07 Feb 2015 | 08 Feb 2015 | 2.16            | 90.19   |       |                          |         |                          |       |         |       |         |
| М    | 00:00 UTC   | 00:00 UTC   | 24.42-          | 123.50- | 0.86  | 0.05±0.004               | 0.19    | 0.02±0.02                | 390.7 | 222.1   | 58.9  | 51.5    |
|      | 18 Feb 2015 | 20 Feb 2015 | 30.05           | 131.79  |       |                          |         |                          |       |         | l     |         |
| N    | 17:00 UTC   | 17:00 UTC   | 34.29-          | 139.89- | -0.27 | -0.03±0.02               | -0.20   | -0.04±0.03               | 478.6 | 220.6   | 53.6  | 52.9    |
|      | 21 Feb 2015 | 23 Feb 2015 | 40.27           | 142.33  |       |                          |         |                          |       |         |       |         |
| 0    | 00.00 01C   | 06 Oct 2015 | 55.90-
60.57 | 192.01- | 0.76  | 0.38±0.05                | 0.38    | 0.09±0.03                | 127.3 | 126.2   | 46.1  | 36.1    |
|      | 14:00 LITC  | 12:00 LITC  | _0.21           | 11/ 13_ |       |                          |         |                          |       |         |       |         |
| Р    | 16 Nov 2015 | 18 Nov 2015 | 0 19            | 119 13  | 0.82  | 0.29±0.03                | 0.83    | 0.37±0.04                | 148.8 | 185.0   | 28.9  | 48.7    |
|      | 18:00 UTC   | 15:00 UTC   | -6 14           | 101.01- |       |                          |         |                          |       |         |       |         |
| Q    | 16 Dec 2015 | 18 Dec 2015 | 4.04            | 102.27  | 0.39  | 0.09±0.04                | -0.75   | -0.03±0.005              | 94.7  | 214.4   | 18.8  | 17.5    |
|      | 04:00 UTC   | 04:00 UTC   | -24.78-         | 110.59- | 0.00  | 0.00.00                  | 0.00    | 0.44+0.00                | 00.7  | 00.4    |       | 07.0    |
| R    | 27 Dec 2015 | 29 Dec 2015 | -21.16          | 112.77  | 0.90  | 0.33±0.03                | 0.89    | 0.44±0.03                | 82.7  | 93.1    | 28.3  | 37.3    |
| 6    | 08:00 UTC   | 20:00 UTC   | 31.97–          | 138.91- | 0.40  | 0.0210.01                | 0.62    | 0.01+0.05                | 257.0 | 160 E   | 46.4  | 46.2    |
| 0    | 23 Jan 2016 | 24 Jan 2016 | 35.03           | 139.94  | -0.40 | -0.02±0.01               | -0.63   | -0.21±0.05               | 357.0 | 168.5   | 46.4  | 40.3    |
| н    | 01:00 UTC   | 01:00 UTC   | 54.78–          | 173.23- | 0.74  | 0.03+0.004               | 0.74    | 0 16+0 02                | 555 7 | 161.4   | 47.0  | 15.3    |
|      | 25 Sep 2016 | 27 Sep 2016 | 61.49           | 184.47  | -0.74 | -0.0310.004              | -0.74   | -0.1010.02               | 333.7 | 101.4   | 47.9  | 45.5    |
| Ш    | 12:00 UTC   | 12:00 UTC   | 47.42-          | 161.78– | 0 00  | 0 00+0 01                | -0 78   | -0 10+0 01               | 317 0 | 139.6   | 43.6  | 36.1    |
| 5    | 27 Sep 2016 | 29 Sep 2016 | 53.85           | 171.57  | 0.00  | 0.00±0.01                | 0.70    | 0.10±0.01                | 017.0 | 100.0   | +0.0  | 00.1    |
| v    | 12:00 UTC   | 10:00 UTC   | 7.66–           | 136.43- | 0.67  | 0.24±0.04                | 0.97    | 97 0.60±0.03             | 92.7  | 99.1    | 25.1  | 34.8    |
|      | 03 Dec 2016 | 05 Dec 2016 | 12.46           | 137.00  |       |                          | 0.01    |                          |       |         |       |         |
| W    | 00:00 UTC   | 00:00 UTC   | 19.86-          | 147.83- | 0.81  | 0.25±0.03                | 0.83    | 0.13+0.02                | 214.8 | 214.6   | 63.0  | 59.0    |
|      | 20 Mar 2017 | 22 Mar 2017 | 27.70           | 152.62  | .     |                          |         |                          | -     | -       |       |         |

**Table 2.** Cases of long-range transport events used for examination of photochemistry (unit for  $\Delta O_3/\Delta CO$  is ppbppbv/ppbppbv).

---

## Author Response (AR2)

**Response to the Co-Editor:**

The authors have successfully addressed the reviewers' comments.
I would like the authors to update me on the status of the papers they refer to that are 'in preparation' and whether they are needed in this manuscript. I find it unsatisfactory to include such references unless absolutely necessary to the arguments presented in the current paper.

We appreciate the Co-Editor's suggestions. Miyazaki et al., in preparation, 2019 is preserved at line 7, page 5, where web url address is given together, to inform the readers that further description of the data set will be available as a paper. Miyazaki et al., in preparation, 2019 at line 10, page 5 is removed. Takashima et al., in preparation, 2019, at line 24, page 13 is removed.

In addition, I have a few minor points that the authors should consider:

Page 1, line 23: '… from the Tropospheric…'

Correction is made.

1, 32: 'obtained by TCR-2.'

Correction is made.

2, 25-23: You have put some of these stations down as GAW and some with the institutional affiliation. They are probably all part of GAW. I suggest you add the affiliations (e.g. NOAA for American Samoa) for those you have listed as GAW. It is more consistent.

For this part, raising all institutions involved to develop the long-term data set is sometimes difficult. Instead, latitudes and longitudes of the sites are listed in a consistent manner in the revised manuscript (page 2, lines 25-32).

4, 4: '…Similar trends were in results from the model members…' I am not sure if that is right, but the current version is a bit unclear.

The sentence is rewritten as 'Similar trends were observed with the ensemble median of model runs of ACCMIP.' (page 4, line 7)

6, 2/3: something is wrong with this sentence. '…exercises because the data were..' ??

Correction is made as suggested.

7, 4: '.. regions where levels less than.'

Correction is made as suggested.

We thank the Co-Editor again for the productive comments.

[revised manuscript text omitted]